# Bayesian Active Learning in the Presence of Nuisance Parameters

**Sabina J. Sloman**[1]         **Ayush Bharti**[2]         **Julien Martinelli**[*3]         **Samuel Kaski**[1,2]

[1]Department of Computer Science, University of Manchester, Manchester, United Kingdom
[2]Department of Computer Science, Aalto University, Helsinki, Finland
[3]Inserm Bordeaux Population Health, Vaccine Research Institute, Université de Bordeaux, Inria Bordeaux Sud-ouest, France

## Abstract

In many settings, such as scientific inference, optimization, and transfer learning, the learner has a well-defined objective, which can be treated as estimation of a *target parameter*, and no intrinsic interest in characterizing the entire data-generating process. Usually, the learner must also contend with additional sources of uncertainty or variables — with *nuisance parameters*. Bayesian active learning, or sequential optimal experimental design, can straightforwardly accommodate the presence of nuisance parameters, and so is a natural active learning framework for such problems. However, the introduction of nuisance parameters can lead to bias in the Bayesian learner's estimate of the target parameters, a phenomenon we refer to as *negative interference*. We characterize the threat of negative interference and how it fundamentally changes the nature of the Bayesian active learner's task. We show that the extent of negative interference can be extremely large, and that accurate estimation of the nuisance parameters is critical to reducing it. The Bayesian active learner is confronted with a dilemma: whether to spend a finite acquisition budget in pursuit of estimation of the target or of the nuisance parameters. Our setting encompasses Bayesian transfer learning as a special case, and our results shed light on the phenomenon of negative transfer between learning environments.

## 1 INTRODUCTION

Sequential optimal experimental design (sOED), or Bayesian active learning, has been used in a wide variety of applications, including mixed-effects modeling [Foster et al., 2019], model selection [Cavagnaro et al., 2010], hy-

perparameter selection [Houlsby et al., 2011], Bayesian optimization [Hernández-Lobato et al., 2014], and graph discovery [Branchini et al., 2023]. In all these applications, the learner prioritizes identification of some aspects of the data-generating process over others: fixed over random effects, functional form/hyperparameter values/location of the function maximum over values of the function's parameters, or graph structure over a structural causal model. In such settings, data is generated by a combination of *target* parameters (which are of interest to the learner) and *nuisance* parameters (which are not). Naïve application of sOED involves greedy maximization of the Bayesian active learner's primary objective — information about the target parameters — marginalized across values of the nuisance parameters.

Despite the frequent use of sOED in such settings, there is little work on how it is affected by the presence of nuisance parameters. As we show in Section 2, this fundamentally changes the nature of the Bayesian active learner's task: while, in the absence of nuisance parameters, the naïve approach usually leads to consistent and sample-efficient estimates [Paninski, 2005], introducing nuisance parameters can result in initial convergence towards the wrong value of the target parameter. We refer to this phenomenon as *negative interference*.

We introduce the concept of negative interference in sOED and provide a non-asymptotic analysis of it. Our results show that (i) negative interference is caused by a bias in the learner's *target* likelihood function that can arise in the presence of nuisance parameters even if their likelihood function is well-specified (Theorem 4.5), (ii) there exist data-generating processes that induce such extreme negative interference that the learner cannot learn from any finite amount of data (Theorem 4.8), and (iii) identification of the (distribution of) nuisance parameters is critical to reducing this threat, and so constitutes an implicit auxiliary objective for the Bayesian active learner (Theorem 4.11). In synthetic but illustrative settings, we show empirically that rather than only occurring in pathological cases, negative

---

[*]Work done while at Aalto University.

interference can arise frequently in practice. Bayesian active learners thus face a dilemma, akin to but distinct from the exploration–exploitation dilemma: should they spend their acquisition budget directly pursuing identification of the target parameters, or of the nuisance parameters?

## 2 PRELIMINARIES

**Notation.** We denote sets by calligraphic capital letters ($\mathscr{A}$) and vectors by bold lowercase letters ($\mathbf{a}$). $\mathbf{a}_i$ is the $i$th entry of $\mathbf{a}$. Bold capital letters ($\mathbf{A}$) indicate a random variable (RV) with a corresponding sample space $\mathscr{A}$. For RVs whose distribution the learner has access to, $P_{\mathbf{A}}$ is the distribution of $\mathbf{A}$ and $p(\mathbf{a})$ is the probability of event $\mathbf{a}$. For RVs whose distribution the learner does not have access to, $Q_{\mathbf{A}}$ is the distribution of $\mathbf{A}$ and $q(\mathbf{a})$ is the probability of event $\mathbf{a}$. $\mathbf{A}$ may refer to the RV with distribution $P_{\mathbf{A}}$ or $Q_{\mathbf{A}}$, and we may not distinguish when it is clear from context.

**General formulation.** On each of several time steps $t \in [1 \dots T]$, the learner selects one action $\mathbf{x}$ from some action set $\mathscr{X}$. Each action results in the realization of an outcome variable that can take values $\mathbf{y} \in \mathscr{Y}$. The outcome distribution is determined by $\mathbf{x}$ and the values of some unobservable parameters, partitioned into *target parameters* that can take values $\boldsymbol{\theta} \in \mathscr{T}$ and *nuisance parameters* that can take values $\boldsymbol{\psi} \in \mathscr{S}$. The probability of observing outcome $\mathbf{y}$ given a tuple $(\mathbf{x}, \boldsymbol{\theta}, \boldsymbol{\psi})$ is $p(\mathbf{y}|\mathbf{x}, \boldsymbol{\theta}, \boldsymbol{\psi})$.

As usual, there is assumed to be some value of the target parameters underlying the data-generating process the learner interacts with, which we denote $\boldsymbol{\theta}^\star$. The learner's goal, which we refer to as their *primary objective*, is identification of the target parameters, i.e., of $\boldsymbol{\theta}^\star$.

There is no assumption placed on the nature of the nuisance parameters underlying the data-generating process. Each data point the learner encounters may be generated by a deterministic value of the nuisance parameters, or by a stochastic value drawn from some distribution. Moreover, we do not require that these draws be independent nor identically distributed: the distribution of the nuisance parameters may change over time, and the learner may or may not have access to the nature of these changes.

Formally, values $\boldsymbol{\psi}$ are assumed to be drawn from a distribution $Q_{\boldsymbol{\Psi}}$, which may depend arbitrarily on the history of observations. The tuple $(\boldsymbol{\theta}^\star, Q_{\boldsymbol{\Psi}})$, which is unknown to the learner, fully specifies a data-generating process (DGP). A DGP $(\boldsymbol{\theta}^\star, Q_{\boldsymbol{\Psi}})$ generates the outcome distribution $Q_{\mathbf{Y}|\mathbf{x}}$, where for any $\mathbf{y}$, $q(\mathbf{y}|\mathbf{x}) = \mathbb{E}_{\boldsymbol{\psi} \sim Q_{\boldsymbol{\Psi}}}[p(\mathbf{y}|\mathbf{x}, \boldsymbol{\theta}^\star, \boldsymbol{\psi})]$. Since the learner does not have access to $(\boldsymbol{\theta}^\star, Q_{\boldsymbol{\Psi}})$, they are not able to evaluate this outcome distribution.

To capture their uncertainty about the value $\boldsymbol{\theta}^\star$ and the distribution $Q_{\boldsymbol{\Psi}}$, the learner considers them RVs with a prior distribution $P_{\boldsymbol{\Theta}, \boldsymbol{\Psi}}$. The learner's predictive distribution

is $P_{\mathbf{Y}|\mathbf{x}}$, under which $p(\mathbf{y}|\mathbf{x}) = \mathbb{E}_{\boldsymbol{\theta}, \boldsymbol{\psi} \sim P_{\boldsymbol{\Theta}, \boldsymbol{\Psi}}}[p(\mathbf{y}|\mathbf{x}, \boldsymbol{\theta}, \boldsymbol{\psi})]$. We use the term *target likelihood function* to refer to $P_{\mathbf{Y}|\mathbf{x}, \boldsymbol{\theta}}$, under which $p(\mathbf{y}|\mathbf{x}, \boldsymbol{\theta}) = \mathbb{E}_{\boldsymbol{\psi} \sim P_{\boldsymbol{\Psi}|\boldsymbol{\theta}}}[p(\mathbf{y}|\mathbf{x}, \boldsymbol{\theta}, \boldsymbol{\psi})]$.

The learner's prior on each time step conditions on the entire history of observations. This results in the posterior $p(\boldsymbol{\theta}|\mathbf{y}, \mathbf{x}) = \frac{p(\mathbf{y}|\mathbf{x}, \boldsymbol{\theta}) \, p(\boldsymbol{\theta})}{p(\mathbf{y}|\mathbf{x})}$ functioning as their prior on the next time step. In general when we refer to the learner's prior $P_{\boldsymbol{\Theta}}$, we suppress this potential dependence on previously-observed data.[1]

The **fixed-$\boldsymbol{\psi}$ formulation** refers to the special case where $Q_{\boldsymbol{\Psi}}$ is degenerate on some value $\boldsymbol{\psi}^\star$ (which is unknown to the learner), and can be written $Q_{\boldsymbol{\Psi}} = \delta(\boldsymbol{\psi}^\star)$, i.e., as the Dirac delta function centered at $\boldsymbol{\psi}^\star$.

**Sequential optimal experimental design (sOED)** is an active learning paradigm in which the learner selects the action $\mathbf{x}$ that maximizes the mutual information (I) between the observations it induces and the value of some specified parameters (see Ryan et al. [2016] or Rainforth et al. [2023] for a review). This quantity is generically known as the expected information gain, which conveys its interpretation as the expectation, across the learner's prior, of the amount of information they gain about the value of the specified parameters. The naïve Bayesian active learner greedily maximizes the expected information gain with respect to the target parameters. We refer to this as the expected target information gain:[2]

**Definition 2.1** (**Expected target information gain (ETIG)**). *The* ETIG *of an action* $\mathbf{x}$ *is the mutual information between* $\mathbf{Y}|\mathbf{x}$ *and* $\boldsymbol{\Theta}$ *according to the learner's prior:*

$$\begin{aligned} \text{ETIG}(\mathbf{x}) &= \text{I}(\boldsymbol{\Theta} \sim P_{\boldsymbol{\Theta}} \; ; \; \mathbf{Y}|\mathbf{x} \sim P_{\mathbf{Y}|\mathbf{x}}) \\ &= \mathbb{E}_{\boldsymbol{\theta} \sim P_{\boldsymbol{\Theta}}}\left[r(\mathbf{x}, \boldsymbol{\theta}, P_{\boldsymbol{\Psi}|\boldsymbol{\theta}})\right] \end{aligned}$$

*where*

$$r(\mathbf{x}, \boldsymbol{\theta}, P_{\boldsymbol{\Psi}|\boldsymbol{\theta}}) \coloneqq \mathbb{E}_{\boldsymbol{\psi} \sim P_{\boldsymbol{\Psi}|\boldsymbol{\theta}}}\left[\mathbb{E}_{\mathbf{y} \sim P_{\mathbf{Y}|\mathbf{x}, \boldsymbol{\theta}, \boldsymbol{\psi}}}\left[\log\left(\frac{p(\mathbf{y}|\mathbf{x}, \boldsymbol{\theta})}{p(\mathbf{y}|\mathbf{x})}\right)\right]\right]$$

*is the learner's* target information gain.

Here, $r(\mathbf{x}, \boldsymbol{\theta}, P_{\boldsymbol{\Psi}|\boldsymbol{\theta}})$ represents the degree to which the learner could expect to achieve their primary objective when taking action $\mathbf{x}$ in DGP $(\boldsymbol{\theta}, P_{\boldsymbol{\Psi}|\boldsymbol{\theta}})$. Of course, the learner does not have access to the true DGP, and so approximates the target information gain they will experience by taking an action $\mathbf{x}$ with respect to their prior $P_{\boldsymbol{\Theta}}$. The ETIG

---

[1] We place no constraints on how the learner's prior over $\boldsymbol{\psi}$ will respond or has responded to data. The learner may similarly update $P_{\boldsymbol{\Psi}}$ on the basis of presently-observed data, and/or $P_{\boldsymbol{\Psi}}$ may implicitly depend on previously-observed data.

[2] We hereafter reserve the term "expected information gain" to refer to the mutual information between observations and the entire data-generating process (see Definition 4.12).

can also be interpreted as a measure of expected posterior concentration, i.e., of the degree to which the learner expects their posterior to concentrate on $\boldsymbol{\theta}^\star$ after observing data $\mathbf{y}|\mathbf{x}$.

While the ETIG may be a useful approximation for the epistemically uncertain learner, the amount of information gained by the learner in practice will depend on the data-generating process. The data does not care about the learner's uncertainty, and will present itself according to the true DGP $(\boldsymbol{\theta}^\star, Q_{\boldsymbol{\Psi}})$. A learner interacting with this DGP will experience a target information gain $r(\mathbf{x}, \boldsymbol{\theta}^\star, Q_{\boldsymbol{\Psi}})$, which we hereafter denote $r^\star(\mathbf{x})$:

$$
\begin{aligned}
r^\star(\mathbf{x}) &\coloneqq r(\mathbf{x}, \boldsymbol{\theta}^\star, Q_{\boldsymbol{\Psi}}) \\
&= \mathop{\mathbb{E}}_{\boldsymbol{\psi} \sim Q_{\boldsymbol{\Psi}}} \left[ \mathop{\mathbb{E}}_{\mathbf{y} \sim P_{\mathbf{Y}|\mathbf{x}, \boldsymbol{\theta}^\star, \boldsymbol{\psi}}} \left[ \log\left( \frac{p(\mathbf{y}|\mathbf{x}, \boldsymbol{\theta}^\star)}{p(\mathbf{y}|\mathbf{x})} \right) \right] \right] \\
&= \mathop{\mathbb{E}}_{\mathbf{y} \sim Q_{\mathbf{Y}|\mathbf{x}}} \left[ \log\left( \frac{p(\mathbf{y}|\mathbf{x}, \boldsymbol{\theta}^\star)}{p(\mathbf{y}|\mathbf{x})} \right) \right]
\end{aligned}
$$

While in practice the learner cannot know $r^\star(\mathbf{x})$, understanding how it behaves facilitates understanding of whether, and under what conditions, the learner's inferences will resemble the true DGP in practice.

**Negative interference.** In the absence of nuisance parameters, the DGP is fully specified by the value of $\boldsymbol{\theta}^\star$. In this case, $r^\star(\mathbf{x})$ can be written without dependence on $Q_{\boldsymbol{\Psi}}$, i.e., as $r^\star(\mathbf{x}) = r(\mathbf{x}, \boldsymbol{\theta}^\star) = \mathbb{E}_{\mathbf{y} \sim P_{\mathbf{Y}|\mathbf{x}, \boldsymbol{\theta}^\star}} \left[ \log\left( \frac{p(\mathbf{y}|\mathbf{x}, \boldsymbol{\theta}^\star)}{p(\mathbf{y}|\mathbf{x})} \right) \right]$.[3] This is also the Kullback-Leibler divergence (KLD) from $P_{\mathbf{Y}|\mathbf{x}, \boldsymbol{\theta}^\star}$ to $P_{\mathbf{Y}|\mathbf{x}}$. Like all KLD measures, $r^\star(\mathbf{x})$ is nonnegative. In other words, in the absence of nuisance parameters, regardless of the DGP the learner encounters, their posterior will move towards $\boldsymbol{\theta}^\star$.

However, the same is not generally true in the presence of nuisance parameters: the distribution over which the expectation is taken, which is generally unavailable to the learner, need not match the distribution whose density is evaluated in the numerator of the log function, which depends only on the learner's prior. In other words, the presence of nuisance parameters poses the threat of biased inference, i.e., that the learner's posterior will move *away* from $\boldsymbol{\theta}^\star$. Such cases correspond to a situation in which the learner is better off *before* learning from data!

When this threat is present, i.e., $r^\star(\mathbf{x})$ is negative due to the presence of nuisance parameters, we say that the learner experiences negative interference (from the nuisance parameters):

**Definition 2.2 (Negative and positive interference).** *If*

$r^\star(\mathbf{x}) < 0$ *for the set of actions* $\mathbf{x}$ *that the learner takes when interacting with DGP* $(\boldsymbol{\theta}^\star, Q_{\boldsymbol{\Psi}})$, *we say the learner experiences negative interference. Otherwise, we say they experience positive interference.*

As a simple example of a setting in which negative interference can occur, consider a learner who confronts data generated by the following linear model:

$$
\mathbf{y}|\mathbf{x} \sim \mathcal{N}\left( \boldsymbol{\theta} x_1 + \boldsymbol{\psi} x_2, \sigma^2 \right) \tag{1}
$$

where the outcome variance $\sigma^2$ is known to the learner. However, the learner may be mistaken about the distribution of $\boldsymbol{\Psi}$. The degree to which they are mistaken will affect their estimate of $\boldsymbol{\theta}$. Informally, if $Q_{\boldsymbol{\Psi}}$ is centered on large values of $\boldsymbol{\psi}$, the learner will tend to observe large values of $\mathbf{y}$. However, if the learner's prior $P_{\boldsymbol{\Psi}}$ is centered on small values of $\boldsymbol{\psi}$, they will attribute the observed large values of $\mathbf{y}$ to a large value of $\boldsymbol{\theta}$. Importantly, this will occur regardless of the true value $\boldsymbol{\theta}^\star$, and if $\boldsymbol{\theta}^\star$ is relatively small, the result will be negative interference.

## 3 RELATED WORK

In Section 4, we characterize the phenomenon of negative interference by analyzing the target information gain $r^\star(\mathbf{x})$. Due to the generality of our formulation, these results apply in a variety of settings, including active model selection, Bayesian optimization, and transfer learning. To further motivate our results, we here summarize how our work complements these and other streams of related work.

**Misspecification and active learning.** As we will show, negative interference arises from a difference between the true nuisance parameter distribution $Q_{\boldsymbol{\Psi}}$ and the learner's prior $P_{\boldsymbol{\Psi}}$, which is a form of prior misspecification. Cuong et al. [2016] and Simchowitz et al. [2021] provide theoretical results on how prior misspecification affects active learning strategies, but do not consider sOED specifically or the presence of nuisance parameters. The results in Simchowitz et al. [2021] apply to a broad range of Bayesian decision-making strategies but in the presence of an exploration–exploitation dilemma, where the learner trades off seeking information with reward maximization. Our results apply in the presence of a trade-off between seeking information about target versus nuisance parameters. Our work is also closely related to Go and Isaac [2022], who propose a novel acquisition function for sOED that facilitates robustness to prior misspecification. Their Robust Expected Information Gain (REIG) is designed to cope with misspecification over the target, rather than nuisance, parameters.[4] We also propose

---

[3]As in the presence of nuisance parameters, $r^\star(\mathbf{x})$ can equivalently be written $r^\star(\mathbf{x}) = \mathbb{E}_{\mathbf{y} \sim Q_{\mathbf{Y}|\mathbf{x}}} \left[ \log\left( \frac{p(\mathbf{y}|\mathbf{x}, \boldsymbol{\theta}^\star)}{p(\mathbf{y}|\mathbf{x})} \right) \right]$ (recall that an action and DGP fully determine the probability of an outcome $p(\mathbf{y}|\mathbf{x}, \boldsymbol{\theta}^\star)$, and so $P_{\mathbf{Y}|\mathbf{x}, \boldsymbol{\theta}^\star} = Q_{\mathbf{Y}|\mathbf{x}, \boldsymbol{\theta}^\star} = Q_{\mathbf{Y}|\mathbf{x}}$).

[4]The REIG can be extended to cope with misspecification of the target likelihood, of which misspecification of the distribution of nuisance parameters is a special case.

an acquisition function which, unlike the REIG, actively seeks information about the nuisance parameters.

Our work is also related to, but distinct from, work on *model misspecification* in active learning [Sugiyama, 2005, Fudenberg et al., 2017, Farquhar et al., 2021, Bogunovic and Krause, 2021, Chen et al., 2021, Sloman et al., 2022, Overstall and McGree, 2022, Catanach and Das, 2023]. Like our setting, model misspecification is characterized by bias in the target likelihood. Unlike our setting, this source of bias is a fixed part of the learner's model, and cannot be alleviated by collecting more data.

**Applications of sOED in the presence of nuisance parameters.** Many applications of sOED involve estimation in the presence of nuisance parameters, or of an embedded model [Foster, 2021]. The effect of prior misspecification has, to a limited degree, been explored in the context of some of these specific applications. To the best of our knowledge we are the first to provide a general characterization of the effect of nuisance parameters.

In **active model selection**, the model indicator is treated as the target parameter and each model's parameters are treated as nuisance parameters [Cavagnaro et al., 2010]. Ray et al. [2013] show that naïve sOED for model selection exhibits suboptimal sample efficiency. Sloman et al. [2023] show that misspecified parameter distributions can lead to biased inferences in initial experimental trials, and discussed the resulting model selection–parameter estimation dilemma.

In **Bayesian optimization** (BO), the target parameter is the location of the data-generating function maximum [Hernández-Lobato et al., 2014, Wang and Jegelka, 2017], and the nuisance parameters correspond to the parameters of the data-generating function. Prior work has investigated the limitations of greedy acquisition functions for BO [González et al., 2016] and shown that identification of the function maximum can be threatened by prior misspecification [Schulz et al., 2016, Bogunovic et al., 2018].

**sOED for implicit likelihoods.** In our setting, the likelihood $p(\mathbf{y}|\mathbf{x}, \boldsymbol{\theta})$ is often not defined analytically and is derived by taking an expectation of $p(\mathbf{y}|\mathbf{x}, \boldsymbol{\theta}, \boldsymbol{\psi})$ over the nuisance parameters $\boldsymbol{\Psi}$. In such cases, the target likelihood constitutes an implicit likelihood [Foster et al., 2019], which connects our setting to work on methods for sOED with implicit likelihoods [Ivanova et al., 2021]. Unlike us, this prior work does not consider the potential for bias in the estimation of the target likelihood (the crucial threat posed by the presence of nuisance parameters). Blau et al. [2022] and Lim et al. [2022]'s proposed methods for sOED with implicit likelihoods incorporate stochasticity in the learner's choice of action, and so explore the design space more thoroughly than naïve sOED methods. The objective to explore the design space is distinct from (although perhaps related to) the specific auxiliary objective that arises in our setting

to learn the distribution of nuisance parameters.

**Inference in the presence of nuisance parameters.** Our results align with a large body of literature on the effect of nuisance parameters on inference more broadly [Neyman and Scott, 1948, Basu, 1977, Dawid, 1980]. We differ from this prior work in our emphasis on the interpretation and characterization of these effects in the context of active learning and the learner's decision of how to allocate their finite acquisition budget.

**Bayesian transfer learning and negative transfer.** Bayesian transfer learning refers to settings in which little or no data is available from the task in which the learner wants to make predictions, and so they rely on learning from data from other, similar tasks [Suder et al., 2023]. This is often formulated as the training and test tasks being characterized by a set of common parameters and a set of task-specific parameters [Suder et al., 2023]. Examples include probabilistic meta-learning [Gordon et al., 2019], Model Agnostic Meta-Learning, where the transferable parameter is a good initialization for training within tasks [Finn et al., 2017, Grant et al., 2018, Yoon et al., 2018, Patacchiola et al., 2020], and multitask learning, where the transferable parameter corresponds to a latent representation shared across tasks [Caruana, 1997].

This formulation can be naturally cast into our setting by considering the transferable parameters to be target parameters, and the task-specific parameters to be nuisance parameters. A DGP can be interpreted to extend across one or many training tasks, and $Q_{\boldsymbol{\Psi}}$ as the distribution of task-specific parameters across these tasks.

Negative transfer refers to the phenomenon that learning in one environment hurts performance in another environment [Wang et al., 2019]. One of the reasons for this can be understood by our more general concept of negative interference. Here, the auxiliary objective can be thought of as learning task-specific properties of the training task(s) that are not expected to transfer to the test tasks. Some Bayesian but non-active transfer learning algorithms do as much by simultaneously learning a fixed, transferable parameter and a distribution over task-specific parameters [Gordon et al., 2019] or explicitly learning a task-specific parameter in each task [Yoon et al., 2018].

# 4 ANALYZING NEGATIVE INTERFERENCE

The primary objective of the Bayesian active learner can be restated as maximizing the amount of positive interference, which requires minimizing the threat of negative interference. To this end, our results address the following ques-

tions:[5] Under what conditions does negative interference arise (Theorem 4.5)? How drastic can the amount of negative interference be (Theorem 4.8)? How can the learner mitigate the threat of negative interference (Theorem 4.11)?

We first give Proposition 4.1, which decomposes $r^\star(\mathbf{x})$ in a way that allows us to make the notion of bias in the target likelihood more rigorous. The derivation of Proposition 4.1 is given in Appendix A.2.

**Proposition 4.1** (Decomposition of $r^\star$). $r^\star(\mathbf{x})$ *can be decomposed as*

$$r^\star(\mathbf{x}) = \underbrace{D_{\mathrm{KL}}\left(Q_{\mathbf{Y}|\mathbf{x}} \,||\, P_{\mathbf{Y}|\mathbf{x}}\right)}_{\mathcal{D}} - \underbrace{D_{\mathrm{KL}}\left(Q_{\mathbf{Y}|\mathbf{x}} \,||\, P_{\mathbf{Y}|\mathbf{x},\boldsymbol{\theta}^\star}\right)}_{\mathcal{D}_{\boldsymbol{\theta}^\star}}.$$

When the true outcome distribution, characterized by the density $q(\mathbf{y}|\mathbf{x})$, differs from the learner's predictive distribution, characterized by the density $p(\mathbf{y}|\mathbf{x})$, the learner's predictive distribution is biased, i.e., does not match the true outcome distribution. We use $\mathcal{D}$ to refer to the extent of this form of bias.

Recall that $q(\mathbf{y}|\mathbf{x}) = \mathbb{E}_{\boldsymbol{\psi} \sim Q_{\boldsymbol{\Psi}}}[p(\mathbf{y}|\mathbf{x},\boldsymbol{\theta}^\star,\boldsymbol{\psi})]$, and that the target likelihood $p(\mathbf{y}|\mathbf{x},\boldsymbol{\theta}^\star) = \mathbb{E}_{\boldsymbol{\psi} \sim P_{\boldsymbol{\Psi}|\boldsymbol{\theta}^\star}}[p(\mathbf{y}|\mathbf{x},\boldsymbol{\theta}^\star,\boldsymbol{\psi})]$. When $q(\mathbf{y}|\mathbf{x})$ differs from $p(\mathbf{y}|\mathbf{x},\boldsymbol{\theta}^\star)$, the target likelihood of $\boldsymbol{\theta}^\star$ is biased, in the sense that the learner's expectations of outcomes delivered under $\boldsymbol{\theta}^\star$ do not resemble the outcomes the world delivers under $\boldsymbol{\theta}^\star$. We use $\mathcal{D}_{\boldsymbol{\theta}^\star}$ to refer to the extent of this form of bias.

Note that both these forms of bias can occur as a result either of epistemic uncertainty about $Q_{\boldsymbol{\Psi}}$ or of model misspecification, i.e., misspecification of the functional form of the likelihood $p(\mathbf{y}|\mathbf{x},\boldsymbol{\theta},\boldsymbol{\psi})$. While our setting allows for only the first source of bias, some elements of our analysis could be extended to understand the effect of model misspecification (see related work in Section 3). We do not consider such extensions further here, but consider them a promising avenue for future work.

**Conditions for negative interference.** Proposition 4.1 provides the intuition that higher bias in the target likelihood leads to a larger extent of negative interference. However, notice that $\mathcal{D}$ also implicitly depends on $\mathcal{D}_{\boldsymbol{\theta}^\star}$: $P_{\mathbf{Y}|\mathbf{x}}$ is constructed by marginalizing over $P_{\mathbf{Y}|\mathbf{x},\boldsymbol{\theta}}$ for all $\boldsymbol{\theta} \in \mathscr{T}$, including $\boldsymbol{\theta}^\star$. Conditions that lead to higher $\mathcal{D}_{\boldsymbol{\theta}^\star}$ could also lead to higher $\mathcal{D}$, and so it is not *a priori* obvious whether bias in the target likelihood is indeed responsible for negative interference. Theorem 4.5 provides an upper bound on $r^\star(\mathbf{x})$ that isolates the contribution of $\mathcal{D}_{\boldsymbol{\theta}^\star}$, and shows that negative interference is a direct function of bias in the target likelihood of $\boldsymbol{\theta}^\star$. It depends on the following definitions:

**Definition 4.2** ($\epsilon$-neighborhood of $\boldsymbol{\theta}$ ($N_\epsilon(\boldsymbol{\theta})$)). $N_\epsilon(\boldsymbol{\theta}) :=$

$\{\tilde{\boldsymbol{\theta}} \in \mathscr{T} \mid d(\boldsymbol{\theta},\tilde{\boldsymbol{\theta}}) < \epsilon\}$, *where $d$ is a suitable distance measure, is the $\epsilon$-neighborhood of $\theta$.*

**Definition 4.3** ($\tilde{P}_{\boldsymbol{\Theta},\mathbf{Y}|\mathbf{x}}$). $\tilde{P}_{\boldsymbol{\Theta},\mathbf{Y}|\mathbf{x}}$ *refers to the joint distribution of $\boldsymbol{\Theta}$ and $\mathbf{Y}|\mathbf{x}$ obtained by "subtracting" the contribution of $N_\epsilon(\boldsymbol{\theta}^\star)$ from the learner's prior. $\tilde{P}_{\boldsymbol{\Theta}}$ is the marginal distribution of $\boldsymbol{\Theta}$ under $\tilde{P}_{\boldsymbol{\Theta},\mathbf{Y}|\mathbf{x}}$, under which*

$$\tilde{p}(\boldsymbol{\theta}) := \frac{p(\boldsymbol{\theta})}{\int_{\mathscr{T} \setminus N_\epsilon(\boldsymbol{\theta}^\star)} p(\boldsymbol{\theta}) \, d\boldsymbol{\theta}} \text{ for any } \boldsymbol{\theta} \in \mathscr{T} \setminus N_\epsilon(\boldsymbol{\theta}^\star)$$

*and $\tilde{P}_{\mathbf{Y}|\mathbf{x}}$ is the marginal distribution of $\mathbf{Y}|\mathbf{x}$ under $\tilde{P}_{\boldsymbol{\Theta},\mathbf{Y}|\mathbf{x}}$, under which*

$$\tilde{p}(\mathbf{y}|\mathbf{x}) := \int_{\mathscr{T} \setminus N_\epsilon(\boldsymbol{\theta}^\star)} p(\mathbf{y}|\mathbf{x},\boldsymbol{\theta}) \, \tilde{p}(\boldsymbol{\theta}) \, d\boldsymbol{\theta} \text{ for any } \mathbf{y} \in \mathscr{Y}.$$

Theorem 4.5 also depends on the following assumption:

**Assumption 4.4** (Smoothness in target parameter space). *There exists some $\epsilon > 0$ such that*

$$\mathbb{E}_{\mathbf{Y} \sim Q_{\mathbf{Y}|\mathbf{x}}}[\log(p(\mathbf{y}|\mathbf{x}))] \geq$$
$$\mathbb{E}_{\mathbf{Y} \sim Q_{\mathbf{Y}|\mathbf{x}}}\left[\left(\int_{N_\epsilon(\boldsymbol{\theta}^\star)} p(\boldsymbol{\theta}) \, d\boldsymbol{\theta}\right) \log(p(\mathbf{y}|\mathbf{x},\boldsymbol{\theta}^\star)) + \right.$$
$$\left. \left(\int_{\mathscr{T} \setminus N_\epsilon(\boldsymbol{\theta}^\star)} p(\boldsymbol{\theta}) \, d\boldsymbol{\theta}\right) \log(\tilde{p}(\mathbf{y}|\mathbf{x}))\right].$$

*Remark.* Assumption 4.4 says that there is some $\epsilon$-neighborhood around $\boldsymbol{\theta}^\star$ inside which it is "safe" to approximate the expectation of $p(\mathbf{y}|\mathbf{x},\boldsymbol{\theta})$ as $p(\mathbf{y}|\mathbf{x},\boldsymbol{\theta}^\star)$, in the sense that the approximation error is not that large.[6] If the target parameter space is continuous and $p(\mathbf{y}|\mathbf{x},\boldsymbol{\theta})$ is smooth near $\boldsymbol{\theta}^\star$, one would expect the approximation error to decrease as $\epsilon$ approaches 0. If the target parameter space is discrete, Assumption 4.4 holds for any $\epsilon < \min\{d(\boldsymbol{\theta}^\star,\boldsymbol{\theta}) \mid \boldsymbol{\theta} \in \mathscr{T} \setminus \{\boldsymbol{\theta}^\star\}\}$, i.e., for any $\epsilon$ less than the smallest distance from $\boldsymbol{\theta}^\star$ to another value of $\boldsymbol{\theta}$ (in which case, $N_\epsilon(\boldsymbol{\theta}^\star) = \{\boldsymbol{\theta}^\star\}$).

Theorem 4.5 gives an upper bound on $r^\star(\mathbf{x})$ in terms of $\mathcal{D}_{\boldsymbol{\theta}^\star}$. The proof is given in Appendix A.3.

**Theorem 4.5** (Upper bound on $r^\star(\mathbf{x})$.). *Given $(\boldsymbol{\theta}^\star, Q_{\boldsymbol{\Psi}})$ and $\epsilon$ that satisfies Assumption 4.4, $r^\star(\mathbf{x})$ is upper-bounded as*

$$r^\star(\mathbf{x}) \leq \left(\int_{\mathscr{T} \setminus N_\epsilon(\boldsymbol{\theta}^\star)} p(\boldsymbol{\theta}) \, d\boldsymbol{\theta}\right)\left(\widetilde{\mathcal{D}} - \mathcal{D}_{\boldsymbol{\theta}^\star}\right)$$

*where $\widetilde{\mathcal{D}} := D_{\mathrm{KL}}\left(Q_{\mathbf{Y}|\mathbf{x}} \,||\, \tilde{P}_{\mathbf{Y}|\mathbf{x}}\right).$*

---

[5]Except when stated otherwise, all our results hold for the general formulation.

[6]More specifically, we require that, on average across observations $\mathbf{y}|\mathbf{x}$, there is some $\epsilon$-neighborhood around $\boldsymbol{\theta}^\star$ inside which the error from approximating the expected likelihood with $p(\mathbf{y}|\mathbf{x},\boldsymbol{\theta}^\star)$ does not close the Jensen gap between the log marginal likelihood across $\mathscr{T}$ and the expectation of the log marginal likelihood both inside and outside $N_\epsilon(\boldsymbol{\theta}^\star)$.

Importantly, $\widetilde{\mathcal{D}}$ does not depend on $\mathcal{D}_{\boldsymbol{\theta}^\star}$. By isolating the contribution of $\mathcal{D}_{\boldsymbol{\theta}^\star}$ to the bound, Theorem 4.5 shows that the bound decreases as a function of $\mathcal{D}_{\boldsymbol{\theta}^\star}$. In other words, the larger $\mathcal{D}_{\boldsymbol{\theta}^\star}$ is, the higher the threat of negative interference is.

**Extent of negative interference.** Theorem 4.8, given below, shows that the amount of negative interference can be arbitrarily extreme. More precisely, it establishes conditions under which the learner might encounter a DGP in which learning for any finite number of time steps would not yield information about $\boldsymbol{\theta}^\star$.

To prove Theorem 4.8, we analyze how $r^\star(x)$ responds to changes in the true distribution of nuisance parameters. For this, we require a measure of the degree to which a scalar-valued function changes with respect to $Q_{\boldsymbol{\Psi}}$. To this end, we consider possible values of $Q_{\boldsymbol{\Psi}}$ to be members of some parametric family of distributions:

**Definition 4.6** (Parameters of $Q_{\boldsymbol{\Psi}}$ ($\boldsymbol{\phi} \in \mathscr{F}$).)**.** *Possible distributions over nuisance parameters can be represented as vectors $\boldsymbol{\phi} \in \mathscr{F}$ where $\mathscr{F}$ is a closed subset of $\mathbb{R}^p$ for some positive integer $p$. $Q_{\boldsymbol{\Psi}}^{\boldsymbol{\phi}}$ is the distribution of nuisance parameters parameterized by $\boldsymbol{\phi}$.*

Values of $\boldsymbol{\phi}$ can be interpreted as vectors parameterizing a family of nuisance parameter distributions (the family of distributions that constitutes possible values of $Q_{\boldsymbol{\Psi}}^{\boldsymbol{\phi}}$). For example, the entire family of Gaussian distributions can be parameterized by a set of vectors $\boldsymbol{\phi} \in \mathbb{R}^2$, with one dimension corresponding to possible values of each of the mean and variance. Definition 4.6 allows $Q_{\boldsymbol{\Psi}}$ to belong to any such family of parametric distributions, including families parameterized by $p$ data points (e.g., possible outputs of a given deterministic algorithm for density estimation of $p$ data points) or that have a discrete domain (for which each element of $\boldsymbol{\phi}$ can represent the — potentially unnormalized — probability of the corresponding domain element).

Since values of $\boldsymbol{\phi}$ are vectors of finite length, we can analyze how a scalar-valued function $f$ responds to changes in $Q_{\boldsymbol{\Psi}}$ via its gradient with respect to $\boldsymbol{\phi}$. We use $\nabla f(\boldsymbol{\phi})$ to refer to the gradient of $f$ at $\boldsymbol{\phi}$. This can also be interpreted as a measure of the robustness of $f$ to changes in $Q_{\boldsymbol{\Psi}}$.

Theorem 4.8 also requires the following assumption:

**Assumption 4.7** (Unboundedness of $\mathscr{F}$)**.** *$\mathscr{F}$ is unbounded in at least one direction, meaning that for some integer $i \in [1, p]$ one of the following conditions holds:*

*(a) $\forall \boldsymbol{\phi} \in \mathscr{F}, \exists \tilde{\boldsymbol{\phi}} \in \mathscr{F}$ for which $\tilde{\phi}_i < \phi_i$*
*(b) $\forall \boldsymbol{\phi} \in \mathscr{F}, \exists \tilde{\boldsymbol{\phi}} \in \mathscr{F}$ for which $\tilde{\phi}_i > \phi_i$*

*Remark.* Assumption 4.7 says that the domain of at least one parameter of the family of nuisance parameter distributions tends to either infinity or negative infinity. Continuing with

the example above in which $\boldsymbol{\phi}$ parameterizes the family of Gaussian distributions, take $\boldsymbol{\phi}_1$ to correspond to possible values of the mean and $\boldsymbol{\phi}_2$ to correspond to possible values of the variance. Possible values of the mean $\boldsymbol{\phi}_1$ consist of all numbers on the real line $\mathbb{R}$. Since $\mathbb{R}$ tends to both positive and negative infinity, both Assumption 4.7(a) and (b) are satisfied for $i = 1$. Possible values of the variance consist of all numbers on the positive real line $\mathbb{R}^+$. Since $\mathbb{R}^+$ is bounded from below by 0 but tends to positive infinity, Assumption 4.7(b) is satisfied for $i = 2$.

Theorem 4.8 establishes conditions under which $r^\star(\mathbf{x})$ is unbounded below with respect to $\boldsymbol{\phi}$. The proof of Theorem 4.8 is given in Appendix A.4. The smaller the amount of posterior mass on $\boldsymbol{\theta}^\star$, the more data the learner will require to recover $\boldsymbol{\theta}^\star$. Theorem 4.8 implies that for any finite number of time steps, there is a distribution of nuisance parameters that would lead to such extreme posterior concentration away from $\boldsymbol{\theta}^\star$ that it could not be unlearned from collecting data on subsequent time steps.

**Theorem 4.8** (Sufficient conditions for no lower bound on $r^\star(\mathbf{x})$)**.** *Given $\boldsymbol{\theta}^\star$, $\mathscr{F}$ and $i$ that satisfies Assumption 4.7, $r^\star(\mathbf{x})$ does not have a finite lower bound (in the sense that $\forall c \in \mathbb{R}$, $\exists \boldsymbol{\phi} \in \mathscr{F}$ such that $r(\mathbf{x}, \boldsymbol{\theta}^\star, Q_{\boldsymbol{\Psi}}^{\boldsymbol{\phi}}) < c$) if one of the following conditions holds:*

*(a) Assumption 4.7(a) holds **and** $\exists \tilde{\boldsymbol{\phi}} \in \mathscr{F}$ such that $\forall \boldsymbol{\phi} \in \mathscr{F}$ for which $\phi_i < \tilde{\phi}_i$, $\nabla \mathcal{D}_{\boldsymbol{\theta}^\star}(\boldsymbol{\phi})$ exists, $\nabla \mathcal{D}(\boldsymbol{\phi})$ exists, and $\nabla \mathcal{D}_{\boldsymbol{\theta}^\star}(\boldsymbol{\phi})_i < \nabla \mathcal{D}(\boldsymbol{\phi})_i - b$*
*(b) Assumption 4.7(b) holds **and** $\exists \tilde{\boldsymbol{\phi}} \in \mathscr{F}$ such that $\forall \boldsymbol{\phi} \in \mathscr{F}$ for which $\phi_i > \tilde{\phi}_i$, $\nabla \mathcal{D}_{\boldsymbol{\theta}^\star}(\boldsymbol{\phi})$ exists, $\nabla \mathcal{D}(\boldsymbol{\phi})$ exists, and $\nabla \mathcal{D}_{\boldsymbol{\theta}^\star}(\boldsymbol{\phi})_i > \nabla \mathcal{D}(\boldsymbol{\phi})_i + b$*

*for some $b \in \mathbb{R}^+$.*

Theorem 4.8 essentially says that catastrophically negative interference occurs when the predictive distribution $P_{\mathbf{Y}|\mathbf{x}}$ is usually more robust than the target likelihood $P_{\mathbf{Y}|\mathbf{x}, \boldsymbol{\theta}^\star}$ to changes in $Q_{\boldsymbol{\Psi}}$.

As a simple example of a setting in which the conditions of Theorem 4.8 are met, consider again the linear model in Equation (1). As we discussed when introducing this example, here, the more mistaken the learner is about the distribution of $\boldsymbol{\Psi}$, the more they will misattribute their observations to the value of $\boldsymbol{\theta}$. If they are arbitrarily mistaken, the amount of negative interference can become arbitrarily large — in other words, $r^\star(\mathbf{x})$ has no lower bound.

To make this more concrete, let's say the learner assigns to both $\boldsymbol{\Theta}$ and $\boldsymbol{\Psi}$ a Gaussian prior, i.e., $P_{\boldsymbol{\Theta}} = \mathcal{N}(\mu_{\boldsymbol{\theta}}, s_{\boldsymbol{\theta}})$ and $P_{\boldsymbol{\Psi}} = \mathcal{N}(\mu_{\boldsymbol{\psi}}, s_{\boldsymbol{\psi}})$. Let's also say that $Q_{\boldsymbol{\Psi}}$ is known to be Gaussian with standard deviation $s_{\boldsymbol{\psi}}$, but need not be centered at $\mu_{\boldsymbol{\psi}}$. As we mentioned in the remark below Assumption 4.7, possible values of the mean of $Q_{\boldsymbol{\Psi}}$ consist of all numbers on the real line $\mathbb{R}$. The family of nuisance parameter distributions is $\{Q_{\boldsymbol{\Psi}}^{\boldsymbol{\phi}} := \mathcal{N}(\boldsymbol{\phi}, s_{\boldsymbol{\psi}}) \mid \boldsymbol{\phi} \in \mathbb{R}\}$.

Since $\mathbb{R}$ is unbounded in both directions, Assumption 4.7(a) and (b) are satisfied for $i = 1$ (i.e., for the first and only dimension of $\mathscr{F}$). The intuition established above can now be stated more formally as "$r^\star(\mathbf{x})$ will decrease with the magnitude of $\phi$ (as it moves further from $\mu_\psi$); since there is no limit to the magnitude of $\phi$, there is no limit to the extent of negative interference." We show in Appendix A.4 that this intuition holds in the sense that this simple case satisfies both conditions of Theorem 4.8.

**The auxiliary objective.** Theorem 4.5 shows that negative interference arises when $\mathcal{D}_{\boldsymbol{\theta}^\star}$ is large. Theorem 4.11, given below, shows that $\mathcal{D}_{\boldsymbol{\theta}^\star}$ is directly related to the degree to which the learner's prior over nuisance parameters is misspecified, i.e., how different it is from $Q_{\boldsymbol{\Psi}}$. It depends on the following definition:

**Definition 4.9.** $Q_{\boldsymbol{\Psi}}$*-mixing. We say a prior $P_{\boldsymbol{\Psi}|\boldsymbol{\theta}^\star}^{\alpha}$ is $Q_{\boldsymbol{\Psi}}$-mixed with a prior $P_{\boldsymbol{\Psi}|\boldsymbol{\theta}^\star}$ at a mixing rate $\alpha \in [0,1]$ when*
$$P_{\boldsymbol{\Psi}|\boldsymbol{\theta}^\star}^{\alpha} = \alpha Q_{\boldsymbol{\Psi}} + (1-\alpha)P_{\boldsymbol{\Psi}|\boldsymbol{\theta}^\star}.$$

A DGP $(\boldsymbol{\theta}^\star, Q_{\boldsymbol{\Psi}})$ and prior $P_{\boldsymbol{\Psi}|\boldsymbol{\theta}^\star}$ define a family of $Q_{\boldsymbol{\Psi}}$-mixed priors whose members are uniquely identified by the $Q_{\boldsymbol{\Psi}}$-mixing rate $\alpha$. We write $P_{\boldsymbol{\Psi}|\boldsymbol{\theta}^\star}^{\alpha}$ for the member of this family $Q_{\boldsymbol{\Psi}}$-mixed at rate $\alpha$, and $\mathcal{D}_{\boldsymbol{\theta}^\star}(\alpha)$ for the value of $\mathcal{D}_{\boldsymbol{\theta}^\star}$ when $P_{\boldsymbol{\Psi}|\boldsymbol{\theta}^\star}^{\alpha}$ is used as a prior and data is generated by the provided DGP. The concept of $Q_{\boldsymbol{\Psi}}$-mixing allows us to directly compare the degree of misspecification of some pairs of priors: given $(\boldsymbol{\theta}^\star, Q_{\boldsymbol{\Psi}})$ and $P_{\boldsymbol{\Psi}|\boldsymbol{\theta}^\star}$, we can say that $P_{\boldsymbol{\Psi}|\boldsymbol{\theta}^\star}^{\alpha_1}$ is more misspecified than $P_{\boldsymbol{\Psi}|\boldsymbol{\theta}^\star}^{\alpha_2}$ if $\alpha_1 < \alpha_2$, in the sense that it differs more from $Q_{\boldsymbol{\Psi}}$.

Theorem 4.11 uses the following lemma, which provides a lower bound on $\mathcal{D}_{\boldsymbol{\theta}^\star}$ as a function of $\alpha$. We refer to this bound, which is also a function of $\alpha$, as $\underline{\mathcal{D}}_{\boldsymbol{\theta}^\star}(\alpha)$. The proof of Lemma 4.10 is given in Appendix A.5.

**Lemma 4.10.** *Lower bound on $\mathcal{D}_{\boldsymbol{\theta}^\star}$. Given $(\boldsymbol{\theta}^\star, Q_{\boldsymbol{\Psi}})$, $P_{\boldsymbol{\Psi}|\boldsymbol{\theta}^\star}$, and $\alpha \in [0,1]$, $\mathcal{D}_{\boldsymbol{\theta}^\star}(\alpha)$ is lower-bounded as*

$$\mathcal{D}_{\boldsymbol{\theta}^\star}(\alpha) \geq \underbrace{-\log\left(\alpha + (1-\alpha)\left(\mathop{\mathbb{E}}_{\mathbf{y}\sim Q_{\mathbf{Y}|\mathbf{x}}}\left[\frac{p(\mathbf{y}|\mathbf{x},\boldsymbol{\theta}^\star)}{q(\mathbf{y}|\mathbf{x})}\right]\right)\right)}_{\underline{\mathcal{D}}_{\boldsymbol{\theta}^\star}(\alpha)}$$

Theorem 4.11 states how this bound depends on $\alpha$, our proxy for the degree of misspecification. The proof is given in Appendix A.5.

**Theorem 4.11** ($\underline{\mathcal{D}}_{\boldsymbol{\theta}^\star}$ *depends on $\alpha$.*). *Given $(\boldsymbol{\theta}^\star, Q_{\boldsymbol{\Psi}})$, $P_{\boldsymbol{\Psi}|\boldsymbol{\theta}^\star}$, $\alpha_1 \in [0,1)$, and $\alpha_2 \in (0,1] > \alpha_1$,*

$$\underline{\mathcal{D}}_{\boldsymbol{\theta}^\star}(\alpha_2) < \underline{\mathcal{D}}_{\boldsymbol{\theta}^\star}(\alpha_1)$$

*if $P_{\boldsymbol{\Psi}|\boldsymbol{\theta}^\star}^{\alpha_1}$ induces negative interference when data is generated from the DGP $(\boldsymbol{\theta}^\star, Q_{\boldsymbol{\Psi}})$.*

Theorem 4.11 shows that in the presence of negative interference, a lower degree of prior misspecification translates to a lower extent of bias in the target likelihood. By Theorem 4.5, this is expected to reduce the extent of negative interference, and so we refer to gathering information about $Q_{\boldsymbol{\Psi}}$ as the learner's *auxiliary objective*.

In the fixed-$\psi$ formulation, Theorem 4.11 has an intuitive interpretation: the threat of negative interference is reduced by effective estimation of $\psi^\star$. This can be seen by substituting $\delta(\psi^\star)$ for $Q_{\boldsymbol{\Psi}}$ and observing that this implies that the bound decreases as the learner places more probability mass on $\psi^\star$. In other words, in this case, the auxiliary objective corresponds to concentration of the learner's posterior over $\boldsymbol{\Psi}|\boldsymbol{\theta}^\star$ onto $\psi^\star$. Leveraging this insight, we define a specific acquisition function for the auxiliary objective in the fixed-$\psi$ formulation.

**Acquisition function for the auxiliary objective.** In the fixed-$\psi$ formulation, the learner's auxiliary objective can be represented in the sOED framework as the learner's expected information gain with respect to $\boldsymbol{\Psi}|\boldsymbol{\theta}^\star$. We define the expected likelihood information gain (ELIG) to stress that this corresponds indirectly to information about the target likelihood of $\boldsymbol{\theta}^\star$. Of course, the learner does not have access to $\boldsymbol{\theta}^\star$, and so the ELIG is defined as an expectation across $P_{\boldsymbol{\Theta}}$.

**Definition 4.12** (**Expected likelihood information gain (ELIG)**). *The ELIG of an action $\mathbf{x}$ is an expectation w.r.t. $P_{\boldsymbol{\Theta}}$ of the mutual information between $\mathbf{Y}|\mathbf{x}$ and $\boldsymbol{\Psi}|\boldsymbol{\theta}$ according to the learner's prior:*

$$\mathrm{ELIG}(\mathbf{x}) = \mathop{\mathbb{E}}_{\boldsymbol{\theta}\sim P_{\boldsymbol{\Theta}}}\left[\mathrm{I}(\boldsymbol{\Psi}|\boldsymbol{\theta}\sim \mathrm{P}_{\boldsymbol{\Psi}|\boldsymbol{\theta}}\ ;\ \mathbf{Y}|\mathbf{x}\sim \mathrm{P}_{\mathbf{Y}|\mathbf{x}})\right]$$
$$= \mathrm{EIG}(\mathbf{x}) - \mathrm{ETIG}(\mathbf{x})$$

*where $\mathrm{EIG}(\mathbf{x}) := \mathrm{I}((\boldsymbol{\Theta},\boldsymbol{\Psi})\sim \mathrm{P}_{\boldsymbol{\Theta},\boldsymbol{\Psi}}\ ;\ \mathbf{Y}|\mathbf{x}\sim \mathrm{P}_{\mathbf{Y}|\mathbf{x}})$ is the* expected information gain (EIG).

The ELIG and EIG subsume, and make explicit the theoretical motivation for, application-specific acquisition functions present in the literature. In the context of model selection, the total entropy function [Borth, 1975] corresponds to the EIG; policies which alternate between acquisition functions tailored to parameter estimation and model selection [Cavagnaro et al., 2016] correspond to alternating between the ELIG and ETIG. In the context of BO, the SCoreBO acquisition function for simultaneous learning of the function maximum and hyperparameters corresponds closely to the EIG [Hvarfner et al., 2023] (although note that the EIG and ELIG would also target information gained about the function itself).

**The active learner's dilemma** — the trade-off the learner faces between pursuit of their primary and auxiliary objectives — is transparent in Definition 4.12. The ELIG, which represents the learner's auxiliary objective, depends

negatively on the ETIG, which represents their primary objective. Actions that lead to gains with respect to the learner's primary objective may reduce or even eliminate opportunities for gains with respect to the auxiliary objective. As a toy example, consider again estimation of the linear model in Equation (1). For an action $\mathbf{x} = [x_1, x_2]$ for which $|x_1| \to \infty$ and $x_2 = 0$, ETIG($\mathbf{x}$) $\to \infty$ and ELIG($\mathbf{x}$) = 0.

Notice that $I(\mathbf{\Psi}|\boldsymbol{\theta}^\star \sim P_{\mathbf{\Psi}|\boldsymbol{\theta}^\star} \; ; \; \mathbf{Y}|\mathbf{x} \sim P_{\mathbf{Y}|\mathbf{x}}) = 0$ i.f.f. $\mathbf{\Psi}|\boldsymbol{\theta}^\star$ and $\mathbf{Y}|\mathbf{x}$ are independent according to the learner's prior. In this case, viewing data $(\mathbf{x}, \mathbf{y})$ has no effect on the learner's posterior on $\mathbf{\Psi}|\boldsymbol{\theta}^\star$ ($P_{\mathbf{\Psi}|\boldsymbol{\theta}^\star, \mathbf{y}, \mathbf{x}} = P_{\mathbf{\Psi}|\boldsymbol{\theta}^\star}$). When this becomes their prior on the next time step, $\mathcal{D}_{\boldsymbol{\theta}^\star}$ remains unchanged. At the same time, $\mathcal{D}$ usually decreases as the learner updates their joint posterior on $(\mathbf{\Theta}, \mathbf{\Psi})$ in light of the data. The upshot is that in this scenario, $r^\star(\mathbf{x})$ decreases (by a simple application of Proposition 4.1). In other words, failure to achieve gains with respect to the auxiliary objective can lead the extent of negative interference to *increase* as the learner collects more data! By directly pursuing their primary objective, the learner risks missing opportunities for gains in their auxiliary objective and the possible exacerbation of negative interference.

## 5 ILLUSTRATING THE PROBLEM

The goal of this section is to illustrate characteristics of learning problems that pose a threat of negative interference and an active learner's dilemma. In three synthetic settings, we demonstrate a substantial threat of negative interference which depends on how misspecified the learner's prior $P_{\mathbf{\Psi}|\boldsymbol{\theta}^\star}$ is with respect to $Q_{\mathbf{\Psi}}$ (corroborating Theorem 4.11) and a trade-off between maximization of the ETIG and ELIG. Further details of the experiments are in Appendix B.

**Illustrative settings.** All of these settings are modeled using the fixed-$\psi$ formulation, i.e., when we refer to a true outcome distribution, it is always for a single value $\psi^\star$.

The first setting is a **linear model**, and corresponds to estimation of a single coefficient in a multiple regression model. This example is intentionally stylized to provide clear intuition. The model is $\mathbf{y} \sim \mathcal{N}(\boldsymbol{\theta}\mathbf{x}_1 + \boldsymbol{\psi}_1\mathbf{x}_2 + \boldsymbol{\psi}_2\mathbf{x}_3 + \boldsymbol{\psi}_3\mathbf{x}_4, \sigma^2)$. The learner's prior specifies that each coefficient is independent and distributed as $\mathcal{N}(0, 10)$. $\mathbf{x}_2$, $\mathbf{x}_3$ and $\mathbf{x}_4$ are positively correlated with each other; $\mathbf{x}_1$, whose effect is the target effect of interest to the learner, is negatively correlated with the inverse of $\mathbf{x}_{(2:4)}$. The extreme multicollinearity induces dependence of the target likelihood on $\psi$.

The second setting, **preference modeling**, corresponds to recovery of a human user's latent preference function on the basis of their observed choices. Our setting is a ver-

sion of an experimental setting from Foster et al. [2019], modified in ways described in Appendix B.2. The model is $\mathbf{y} \sim \text{Bernoulli}\left(\left(1 + e^{\boldsymbol{\psi}\mathbf{x} - \boldsymbol{\theta}}\right)^{-1}\right)$. The learner's prior specifies $\mathbf{\Theta}$ and $\mathbf{\Psi}$ as independent Gaussian RVs, $P_{\mathbf{\Theta}} = \mathcal{N}(0, 16)$ and $P_{\mathbf{\Psi}} = \mathcal{N}(0, 1)$. The target parameter $\boldsymbol{\theta}$ may correspond to a stable preference that generalizes across users or settings, and $\psi$ may correspond to the accuracy with which a user encodes the choice set in a given setting. Past research has shown that in similar models, interdependency between $\boldsymbol{\theta}$ and $\psi$ is induced by the structure of the model [Krefeld-Schwalb et al., 2022].

The final setting is **Gaussian Process (GP) regression** [Rasmussen and Williams, 2006]. The GP prior has a zero mean with covariance determined by a composite kernel $k(\cdot, \cdot) = k_{\boldsymbol{\theta}}(\cdot, \cdot) + k_{\boldsymbol{\psi}_1}(\cdot, \cdot)$, where $\boldsymbol{\theta}$ corresponds to the lengthscale of $k_{\boldsymbol{\theta}}(\cdot, \cdot)$ and $\boldsymbol{\psi}_1$ corresponds to the lengthscale of $k_{\boldsymbol{\psi}_1}(\cdot, \cdot)$. The learner's prior specified $P_{\mathbf{\Theta}}$ and $P_{\mathbf{\Psi}_1}$ to be independent Gamma distributions, both with concentration 3 and scale 1.25. Interdependency between $\boldsymbol{\theta}$ and $\boldsymbol{\psi}_1$ is induced by their composite effect on the correlation structure of the observed data. The lengthscale parameter $\boldsymbol{\psi}_1$ is not the only relevant nuisance parameter, however: the function sampled from the GP defined by $k(\cdot, \cdot)$ ultimately determines the outcome distribution, and so we additionally model it as a nuisance parameter and refer to it as $\boldsymbol{\psi}_2$.

**Prior misspecification leads to more negative interference.** Figures 1a, 1c and 1e plot $r^\star(\mathbf{x})$ as a function of $p(\boldsymbol{\psi}^\star|\boldsymbol{\theta}^\star)$ (higher values on the $x$-axis indicate lower misspecification). The plotted values of $\boldsymbol{\theta}$ and $\psi$ are sampled from the learner's prior. These figures yield two insights. In each of these illustrative settings, a substantial proportion of the considered DGPs (i.e., unique values of $(\boldsymbol{\theta}^\star, \boldsymbol{\psi}^\star)$) induce negative interference. In Figure 1a, 42% of the plotted points show DGPs that induce negative interference; this is true for 25% of the plotted points in Figure 1c, and 58% of the plotted points in Figure 1e (the additional complexity of the GP regression setting likely introduces comparatively larger Monte Carlo estimation errors; see Appendix B.3). Secondly, in the presence of negative interference (orange points), $r^\star(\mathbf{x})$ generally increases with $p(\boldsymbol{\psi}^\star|\boldsymbol{\theta}^\star)$, which reflects the result from Theorem 4.11.

**The active learner's dilemma.** Figures 1b, 1d and 1f show how the values of the acquisition functions corresponding to the learner's primary and auxiliary objectives (ETIG and ELIG, respectively) compare to each other. In all cases, there is a trade-off: maximizing with respect to one objective means forgoing gains on the other. For example, Figure 1b shows that in the linear model setting, the ETIG is high when the magnitude of $\mathbf{x}_1$ is large and the magnitude of $\mathbf{x}_{(2:4)}$ is small; this facilitates identification of the effect of $\mathbf{x}_1$. ELIG favors the opposite situation, precisely because it aims for identification of the effects of $\mathbf{x}_{(2:4)}$. Figure 1f shows that in the GP regression setting, the ETIG is maxi-

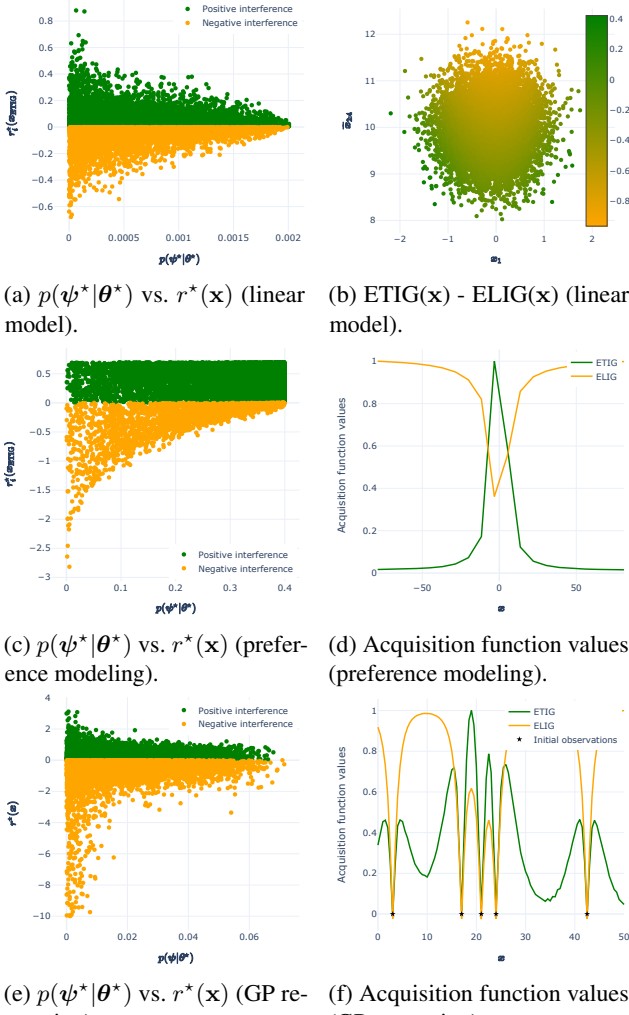

(a) $p(\boldsymbol{\psi}^\star|\boldsymbol{\theta}^\star)$ vs. $r^\star(\mathbf{x})$ (linear model).

(b) ETIG($\mathbf{x}$) - ELIG($\mathbf{x}$) (linear model).

(c) $p(\boldsymbol{\psi}^\star|\boldsymbol{\theta}^\star)$ vs. $r^\star(\mathbf{x})$ (preference modeling).

(d) Acquisition function values (preference modeling).

(e) $p(\boldsymbol{\psi}^\star|\boldsymbol{\theta}^\star)$ vs. $r^\star(\mathbf{x})$ (GP regression).

(f) Acquisition function values (GP regression).

Figure 1: *Notes.* Panels (a,c,e) show 10,000 parameter values drawn from the learner's prior. In panels (a,c), $r^\star$ is computed for the value of $\mathbf{x}$ that maximizes the ETIG. In panel (e), $r^\star$ is computed for two nearby points in the interior of the domain, and values on the $y$-axis are truncated from below at -10. Panels (b,d,f) show the acquisition function values normalized by their respective maximum values. In panel (b), the acquisition function values additionally each have their respective minimum values subtracted (so their minima coincide at 0), and the $y$-axis is $\frac{\mathbf{x}_1+\mathbf{x}_2+\mathbf{x}_3}{3}$. Panel (f) additionally shows initial training points on which the GP prior used to compute the acquisition function values is conditioned.

mized at points near the training data. Different values of $\boldsymbol{\theta}$ considered by the learner's prior make different predictions about how correlated these points should be with the training data (while most values of $\boldsymbol{\theta}$ predict that points further away will revert to the prior mean). By contrast, ELIG focuses its energy on gaining information about the function as a whole, and so prefers points further away from the training

data, where it is most uncertain about the outcomes. See Appendix B.2 for discussion of the preference modeling setting.

# 6 CONCLUSION

We analyzed the phenomenon of negative interference, a threat to Bayesian inference posed by the presence of nuisance parameters, and its effect on Bayesian active learning. Our analysis showed that mitigating the threat of negative interference requires the Bayesian active learner to take into account an auxiliary objective: identification of the distribution of nuisance parameters.

A limitation of our analysis is the assumption that the learner has access to the likelihood function. Extensions of our work could apply components of our analysis to the more general setting of potential model misspecification. Our analysis is also limited in the assumption that the partition between target and nuisance parameters is well-defined; in some settings, the learner may additionally be tasked with learning this partition (which can be thought of as a form of relevant feature selection), and/or have the option to eliminate effects of the nuisance parameters (e.g., by controlling laboratory conditions).

By identifying the threat of negative interference and establishing the theoretical groundwork for how to address it (identification of the distribution of nuisance parameters), our work opens the door to the development of algorithms to navigate the active learner's dilemma. As a step in this direction, Definition 4.12 proposes the ELIG, a novel acquisition function for the auxiliary objective in the fixed-$\psi$ formulation. Applications of sOED in this special case can address the threat of negative interference with design policies that alternate between the ELIG and ETIG according to a principled switching criterion (one that would ideally be sensitive to the extent of the threat of negative interference). This criterion could be based on one of the many existing criteria developed in the context of the exploration–exploitation dilemma (e.g., an $\epsilon$-greedy scheme), the value of a distributionally robust information gain measure [Go and Isaac, 2022], or a credible lower bound of target information gain values.

**Author Contributions**

S. J. Sloman contributed the initial problem formulation, mathematical results and the code for the linear model and preference modeling experiments, and wrote the first draft of the paper. A. Bharti contributed feedback on the problem formulation and mathematical results, and revisions of the paper. J. Martinelli contributed the code for the GP regression experiment and revisions of the paper. S. Kaski contributed feedback on the problem formulation and revi-

sions of the paper.

## ACKNOWLEDGEMENTS

The authors thank Stephen Menary, Thomas Quilter, and Leila Sloman for helpful discussions. SJS and SK were supported by the UKRI Turing AI World-Leading Researcher Fellowship, [EP/W002973/1]. AB was supported by the Academy of Finland Flagship programme: Finnish Center for Artificial Intelligence FCAI. JM acknowledges the support of the Research Council of Finland under the HEALED project (grant 13342077).

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

# Bayesian Active Learning in the Presence of Nuisance Parameters (Supplementary Material)

Sabina J. Sloman[1]        Ayush Bharti[2]        Julien Martinelli[*3]        Samuel Kaski[1,2]

[1]Department of Computer Science, University of Manchester, Manchester, United Kingdom
[2]Department of Computer Science, Aalto University, Helsinki, Finland
[3]Inserm Bordeaux Population Health, Vaccine Research Institute, Université de Bordeaux, Inria Bordeaux Sud-ouest, France

The supplementary material is organized as follows:

- In Appendix A, we provide proofs of all our mathematical results.
- In Appendix B, we provide details of the illustrative examples presented in Section 5.

## A   PROOFS OF MATHEMATICAL RESULTS

### A.1   DEFINITIONS

- $D_{KL}(Q \mid\mid P)$ is the Kullback-Leibler divergence from $Q$ to $P$:

$$D_{KL}(Q \mid\mid P) = \int_{\mathscr{Y}} \log\left(\frac{q(\mathbf{y})}{p(\mathbf{y})}\right) q(\mathbf{y}) \, d\mathbf{y}$$

- $H(Q \mid\mid P)$ is the cross-entropy from $Q$ to $P$:

$$H(Q \mid\mid P) = -\int_{\mathscr{Y}} \log(p(\mathbf{y})) q(\mathbf{y}) \, d\mathbf{y}$$

- $H(Q)$ is the entropy of distribution $Q$:

$$H(Q) = -\int_{\mathscr{Y}} \log(q(\mathbf{y})) q(\mathbf{y}) \, d\mathbf{y}$$

### A.2   DERIVATION OF PROPOSITION 4.1

$$
\begin{aligned}
r^{\star}(\mathbf{x}) &= \mathop{\mathbb{E}}_{\mathbf{y} \sim Q_{\mathbf{Y}|\mathbf{x}}} \left[ \log\left(\frac{p(\mathbf{y}|\mathbf{x}, \boldsymbol{\theta}^{\star})}{p(\mathbf{y}|\mathbf{x})}\right) \right] \\
&= H\left(Q_{\mathbf{Y}|\mathbf{x}} \mid\mid P_{\mathbf{Y}|\mathbf{x}}\right) - H\left(Q_{\mathbf{Y}|\mathbf{x}} \mid\mid P_{\mathbf{Y}|\mathbf{x}, \boldsymbol{\theta}^{\star}}\right) \\
&= H\left(Q_{\mathbf{Y}|\mathbf{x}}\right) + D_{KL}\left(Q_{\mathbf{Y}|\mathbf{x}} \mid\mid P_{\mathbf{Y}|\mathbf{x}}\right) - H\left(Q_{\mathbf{Y}|\mathbf{x}}\right) - D_{KL}\left(Q_{\mathbf{Y}|\mathbf{x}} \mid\mid P_{\mathbf{Y}|\mathbf{x}, \boldsymbol{\theta}^{\star}}\right) \\
&= D_{KL}\left(Q_{\mathbf{Y}|\mathbf{x}} \mid\mid P_{\mathbf{Y}|\mathbf{x}}\right) - D_{KL}\left(Q_{\mathbf{Y}|\mathbf{x}} \mid\mid P_{\mathbf{Y}|\mathbf{x}, \boldsymbol{\theta}^{\star}}\right)
\end{aligned}
$$

---

[*]Work done while at Aalto University.

## A.3 PROOF OF THEOREM 4.5

The proof of Theorem 4.5 depends on Lemma A.1, which upper bounds $\mathcal{D}$ as a function of $\mathcal{D}_{\boldsymbol{\theta}^\star}$.

**Lemma A.1** (Upper bound of $\mathcal{D}$.). *Given $(\boldsymbol{\theta}^\star, Q_{\boldsymbol{\Psi}})$ and $\epsilon$ that satisfies Assumption 4.4, $\mathcal{D}$ is upper-bounded as*

$$\mathcal{D} \leq \left( \int_{N_\epsilon(\boldsymbol{\theta}^\star)} p(\boldsymbol{\theta})\, d\boldsymbol{\theta} \right) \mathcal{D}_{\boldsymbol{\theta}^\star} + \left( \int_{\mathscr{T} \setminus N_\epsilon(\boldsymbol{\theta}^\star)} p(\boldsymbol{\theta})\, d\boldsymbol{\theta} \right) \widetilde{\mathcal{D}}$$

*Proof.*

$$\mathcal{D} = \mathrm{D}_{\mathrm{KL}} \left( Q_{\mathbf{Y}|\mathbf{x}} \,\|\, P_{\mathbf{Y}|\mathbf{x}} \right) \qquad \text{(Proposition 4.1)}$$

$$= \int_{\mathscr{Y}} \log \left( \frac{q(\mathbf{y}|\mathbf{x})}{p(\mathbf{y}|\mathbf{x})} \right) q(\mathbf{y}|\mathbf{x})\, d\mathbf{y}$$

$$= \int_{\mathscr{Y}} \left( \log \left( q(\mathbf{y}|\mathbf{x}) \right) - \log \left( p(\mathbf{y}|\mathbf{x}) \right) \right) q(\mathbf{y}|\mathbf{x})\, d\mathbf{y}$$

$$\leq \int_{\mathscr{Y}} \left( \log \left( q(\mathbf{y}|\mathbf{x}) \right) - \left( \int_{N_\epsilon(\boldsymbol{\theta}^\star)} p(\boldsymbol{\theta})\, d\boldsymbol{\theta} \right) \log \left( p(\mathbf{y}|\mathbf{x}, \boldsymbol{\theta}^\star) \right) - \left( \int_{\mathscr{T} \setminus N_\epsilon(\boldsymbol{\theta}^\star)} p(\boldsymbol{\theta})\, d\boldsymbol{\theta} \right) \log \left( \tilde{p}\left( \mathbf{y}|\mathbf{x} \right) \right) \right) q(\mathbf{y}|\mathbf{x})\, d\mathbf{y}$$

$$\qquad \text{(Assumption 4.4)}$$

$$= -\mathrm{H}\left( Q_{\mathbf{Y}|\mathbf{x}} \right) + \left( \int_{N_\epsilon(\boldsymbol{\theta}^\star)} p(\boldsymbol{\theta})\, d\boldsymbol{\theta} \right) \mathrm{H}\left( Q_{\mathbf{Y}|\mathbf{x}} \,\|\, P_{\mathbf{Y}|\mathbf{x}, \boldsymbol{\theta}^\star} \right) + \left( \int_{\mathscr{T} \setminus N_\epsilon(\boldsymbol{\theta}^\star)} p(\boldsymbol{\theta})\, d\boldsymbol{\theta} \right) \mathrm{H}\left( Q_{\mathbf{Y}|\mathbf{x}} \,\|\, \tilde{P}_{\mathbf{Y}|\mathbf{x}} \right)$$

$$= -\mathrm{H}\left( Q_{\mathbf{Y}|\mathbf{x}} \right) + \left( \int_{N_\epsilon(\boldsymbol{\theta}^\star)} p(\boldsymbol{\theta})\, d\boldsymbol{\theta} \right) \left( \mathrm{H}\left( Q_{\mathbf{Y}|\mathbf{x}} \right) + \mathrm{D}_{\mathrm{KL}}\left( Q_{\mathbf{Y}|\mathbf{x}} \,\|\, P_{\mathbf{Y}|\mathbf{x}, \boldsymbol{\theta}^\star} \right) \right)$$

$$\quad + \left( \int_{\mathscr{T} \setminus N_\epsilon(\boldsymbol{\theta}^\star)} p(\boldsymbol{\theta})\, d\boldsymbol{\theta} \right) \left( \mathrm{H}\left( Q_{\mathbf{Y}|\mathbf{x}} \right) + \mathrm{D}_{\mathrm{KL}}\left( Q_{\mathbf{Y}|\mathbf{x}} \,\|\, \tilde{P}_{\mathbf{Y}|\mathbf{x}} \right) \right)$$

$$= \left( \int_{N_\epsilon(\boldsymbol{\theta}^\star)} p(\boldsymbol{\theta})\, d\boldsymbol{\theta} \right) \mathrm{D}_{\mathrm{KL}}\left( Q_{\mathbf{Y}|\mathbf{x}} \,\|\, P_{\mathbf{Y}|\mathbf{x}, \boldsymbol{\theta}^\star} \right) + \left( \int_{\mathscr{T} \setminus N_\epsilon(\boldsymbol{\theta}^\star)} p(\boldsymbol{\theta})\, d\boldsymbol{\theta} \right) \mathrm{D}_{\mathrm{KL}}\left( Q_{\mathbf{Y}|\mathbf{x}} \,\|\, \tilde{P}_{\mathbf{Y}|\mathbf{x}} \right)$$

$$= \left( \int_{N_\epsilon(\boldsymbol{\theta}^\star)} p(\boldsymbol{\theta})\, d\boldsymbol{\theta} \right) \mathcal{D}_{\boldsymbol{\theta}^\star} + \left( \int_{\mathscr{T} \setminus N_\epsilon(\boldsymbol{\theta}^\star)} p(\boldsymbol{\theta})\, d\boldsymbol{\theta} \right) \widetilde{\mathcal{D}}$$

$\square$

Direct substitution of this bound into Proposition 4.1 completes the proof of Theorem 4.5:

$$r^\star(\mathbf{x}) = \mathcal{D} + \mathcal{D}_{\boldsymbol{\theta}^\star} \qquad \text{(Proposition 4.1)}$$

$$\leq \left( \int_{N_\epsilon(\boldsymbol{\theta}^\star)} p(\boldsymbol{\theta})\, d\boldsymbol{\theta} \right) \mathcal{D}_{\boldsymbol{\theta}^\star} + \left( \int_{\mathscr{T} \setminus N_\epsilon(\boldsymbol{\theta}^\star)} p(\boldsymbol{\theta})\, d\boldsymbol{\theta} \right) \widetilde{\mathcal{D}} - \mathcal{D}_{\boldsymbol{\theta}^\star} \qquad \text{(Lemma A.1)}$$

$$= \left( \int_{\mathscr{T} \setminus N_\epsilon(\boldsymbol{\theta}^\star)} p(\boldsymbol{\theta})\, d\boldsymbol{\theta} \right) \left( \widetilde{\mathcal{D}} - \mathcal{D}_{\boldsymbol{\theta}^\star} \right)$$

## A.4 PROOF OF THEOREM 4.8

Given $i$ that satisfies Assumption 4.7, the conditions of Theorem 4.8 imply that $r^\star(\mathbf{x})$ can grow arbitrarily negative as $\phi_i \to \infty$ or as $\phi_i \to -\infty$.

If $i$ satisfies Assumption 4.7(a), the conditions of Theorem 4.8 imply that at some point $\tilde{\phi}_i$ on dimension $i$ of $\mathscr{F}$, any movement along this dimension towards $-\infty$ will result in $r^\star(\mathbf{x})$ decreasing by at least some amount $b$. Assumption 4.7(a) implies that $\phi_i \to -\infty$, and the stated conditions imply that $r^\star(\mathbf{x})$ can grow arbitrarily negative as $\phi_i \to -\infty$.

More specifically, we require that, for some $\tilde{\phi}$, the following holds $\forall \phi \in \mathscr{F}$ for which $\phi_i < \tilde{\phi}_i$:

$$\nabla r^\star(\mathbf{x})_i > b \iff \nabla \mathcal{D}(\mathbf{x})_i - \nabla \mathcal{D}_{\boldsymbol{\theta}^\star}(\mathbf{x})_i > b$$
$$\iff \nabla \mathcal{D}_{\boldsymbol{\theta}^\star}(\mathbf{x})_i < \nabla \mathcal{D}(\mathbf{x})_i - b$$

as stated in the theorem.

If $i$ satisfies Assumption 4.7(b), the conditions of Theorem 4.8 imply that at some point $\tilde{\phi}_i$ on dimension $i$ of $\mathscr{F}$, any movement along this dimension towards $\infty$ will result in $r^\star(\mathbf{x})$ decreasing by at least some amount $b$. Assumption 4.7(b) implies that $\phi_i \to \infty$, and the stated conditions imply that $r^\star(\mathbf{x})$ can grow arbitrarily negative as $\phi_i \to \infty$.

For this, we require that, for some $\tilde{\phi}$, the following holds $\forall \phi \in \mathscr{F}$ for which $\phi_i > \tilde{\phi}_i$:

$$\nabla r^\star(\mathbf{x})_i < -b \iff \nabla \mathcal{D}(\mathbf{x})_i - \nabla \mathcal{D}_{\boldsymbol{\theta}^\star}(\mathbf{x})_i < -b$$
$$\iff \nabla \mathcal{D}_{\boldsymbol{\theta}^\star}(\mathbf{x})_i > \nabla \mathcal{D}(\mathbf{x})_i + b$$

as stated in the theorem.

**Application to the linear model.** We write the learner's prior over $(\boldsymbol{\Theta}, \boldsymbol{\Psi})$ as $P_{\boldsymbol{\Theta}, \boldsymbol{\Psi}} = \mathcal{N}(\boldsymbol{\mu}, \boldsymbol{\Sigma})$. We use $\sigma^2_{\mathbf{y}|\mathbf{x}}$ and $\sigma^2_{\mathbf{y}|\mathbf{x},\boldsymbol{\theta}^\star}$ to refer to the variance of the learner's predictive distribution and variance of distribution corresponding to the learner's target likelihood, respectively. These are:

$$\sigma^2_{\mathbf{y}|\mathbf{x}} = \sigma^2 + \mathbf{x}\boldsymbol{\Sigma}\mathbf{x}^T \tag{2}$$

$$\sigma^2_{\mathbf{y}|\mathbf{x},\boldsymbol{\theta}^\star} = \sigma^2 + \mathbf{x} \begin{bmatrix} 0 & 0 \\ 0 & \boldsymbol{\Sigma}_{2,2} - \boldsymbol{\Sigma}_{1,2}\boldsymbol{\Sigma}_{1,1}^{-1}\boldsymbol{\Sigma}_{1,2} \end{bmatrix} \mathbf{x}^T \tag{3}$$

Without loss of generality, take $\boldsymbol{\mu} = [0,0]$ and $\boldsymbol{\Sigma}$ to be a diagonal matrix, and so $\mathbb{E}_{\boldsymbol{\theta} \sim P_{\boldsymbol{\Theta}}}[\boldsymbol{\theta}] = 0$ and $\mathbb{E}_{\boldsymbol{\psi} \sim P_{\boldsymbol{\Psi}|\boldsymbol{\theta}^\star}}[\boldsymbol{\psi}] = \mathbb{E}_{\boldsymbol{\psi} \sim P_{\boldsymbol{\Psi}}}[\boldsymbol{\psi}] = 0$.

As described in the main text, $\phi$ represents the mean of $Q_{\boldsymbol{\Psi}}$, i.e., $\phi \in \mathbb{R} := \mathbb{E}_{\boldsymbol{\psi} \sim Q_{\boldsymbol{\Psi}}}[\boldsymbol{\psi}]$. This satisfies both Assumption 4.7(a) and Assumption 4.7(b) (for $i = 1$). Satisfying the conditions of the theorem requires showing that the conditions on the gradients $\nabla \mathcal{D}_{\boldsymbol{\theta}^\star}$ and $\nabla \mathcal{D}$ are met. Below, we give the derivatives $\frac{\partial \mathcal{D}_{\boldsymbol{\theta}^\star}}{\partial \phi}$ and $\frac{\partial \mathcal{D}}{\partial \phi}$, and then show that there is a value of $\tilde{\phi}$ below which the gradient conditions in Theorem 4.8(a) hold, and a value of $\tilde{\phi}$ above which the gradient conditions in Theorem 4.8(b) hold.

$$\mathcal{D}_{\boldsymbol{\theta}^\star} = \mathrm{D}_{\mathrm{KL}}\left(Q_{\mathbf{Y}|\mathbf{x}} \,\|\, P_{\mathbf{Y}|\mathbf{x},\boldsymbol{\theta}^\star}\right)$$
$$= \frac{1}{2}\left(\left(\frac{\sigma^2}{\sigma^2_{\mathbf{y}|\mathbf{x},\boldsymbol{\theta}^\star}}\right) + \frac{\left(\boldsymbol{\theta}^\star x_1 + \mathbb{E}_{\boldsymbol{\psi} \sim P_{\boldsymbol{\Psi}|\boldsymbol{\theta}^\star}}[\boldsymbol{\psi}x_2] - (\boldsymbol{\theta}^\star x_1 + \mathbb{E}_{\boldsymbol{\psi} \sim Q_{\boldsymbol{\Psi}}}[\boldsymbol{\psi}x_2])\right)^2}{\sigma^2_{\mathbf{y}|\mathbf{x},\boldsymbol{\theta}^\star}} - 1 + \log\left(\frac{\sigma^2_{\mathbf{y}|\mathbf{x},\boldsymbol{\theta}^\star}}{\sigma^2}\right)\right)$$
$$= \frac{1}{2}\left(\left(\frac{\sigma^2}{\sigma^2_{\mathbf{y}|\mathbf{x},\boldsymbol{\theta}^\star}}\right) + \frac{(\phi x_2)^2}{\sigma^2_{\mathbf{y}|\mathbf{x},\boldsymbol{\theta}^\star}} - 1 + \log\left(\frac{\sigma^2_{\mathbf{y}|\mathbf{x},\boldsymbol{\theta}^\star}}{\sigma^2}\right)\right)$$
$$\frac{\partial \mathcal{D}_{\boldsymbol{\theta}^\star}}{\partial \phi} = \frac{\phi x_2^2}{\sigma^2_{\mathbf{y}|\mathbf{x},\boldsymbol{\theta}^\star}}$$

$$\mathcal{D} = \mathrm{D}_{\mathrm{KL}}\left(Q_{\mathbf{Y}|\mathbf{x}} \,\|\, P_{\mathbf{Y}|\mathbf{x}}\right)$$
$$= \frac{1}{2}\left(\left(\frac{\sigma^2}{\sigma^2_{\mathbf{y}|\mathbf{x}}}\right) + \frac{\left(\mathbb{E}_{\boldsymbol{\theta},\boldsymbol{\psi} \sim P_{\boldsymbol{\Theta},\boldsymbol{\Psi}}}[\boldsymbol{\theta}x_1 + \boldsymbol{\psi}x_2] - (\boldsymbol{\theta}^\star x_1 + \mathbb{E}_{\boldsymbol{\psi} \sim Q_{\boldsymbol{\Psi}}}[\boldsymbol{\psi}x_2])\right)^2}{\sigma^2_{\mathbf{y}|\mathbf{x}}} - 1 + \log\left(\frac{\sigma^2_{\mathbf{y}|\mathbf{x}}}{\sigma^2}\right)\right)$$
$$= \frac{1}{2}\left(\left(\frac{\sigma^2}{\sigma^2_{\mathbf{y}|\mathbf{x}}}\right) + \frac{(\boldsymbol{\theta}^\star x_1 + \phi x_2)^2}{\sigma^2_{\mathbf{y}|\mathbf{x}}} - 1 + \log\left(\frac{\sigma^2_{\mathbf{y}|\mathbf{x}}}{\sigma^2}\right)\right)$$
$$\frac{\partial \mathcal{D}}{\partial \phi} = \frac{\boldsymbol{\theta}^\star x_1 x_2 + \phi x_2^2}{\sigma^2_{\mathbf{y}|\mathbf{x}}}$$

$$\frac{\partial \mathcal{D}_{\boldsymbol{\theta}^\star}}{\partial \phi} - \frac{\partial \mathcal{D}}{\partial \phi} = \frac{\phi x_2^2}{\sigma_{\mathbf{y}|\mathbf{x},\boldsymbol{\theta}^\star}^2} - \frac{\boldsymbol{\theta}^\star x_1 x_2 + \phi x_2^2}{\sigma_{\mathbf{y}|\mathbf{x}}^2}$$

$$= \phi \left( \frac{x_2^2}{\sigma_{\mathbf{y}|\mathbf{x},\boldsymbol{\theta}^\star}^2} - \frac{x_2^2}{\sigma_{\mathbf{y}|\mathbf{x}}^2} \right) - \frac{\boldsymbol{\theta}^\star x_1 x_2}{\sigma_{\mathbf{y}|\mathbf{x}}^2} \tag{4}$$

Taking $\rho = \frac{x_2^2}{\sigma_{\mathbf{y}|\mathbf{x},\boldsymbol{\theta}^\star}^2} - \frac{x_2^2}{\sigma_{\mathbf{y}|\mathbf{x}}^2} \geq 0$ (since $\sigma_{\mathbf{y}|\mathbf{x},\boldsymbol{\theta}^\star}^2$ can never be greater than $\sigma_{\mathbf{y}|\mathbf{x}}^2$) and $\tau = \frac{\boldsymbol{\theta}^\star x_1 x_2}{\sigma_{\mathbf{y}|\mathbf{x}}^2}$, Equation (4) is at most $-b$ when $\phi \leq \frac{\tau - b}{\rho}$, and so the conditions in Theorem 4.8(a) are met for any $\tilde{\phi} \leq \frac{\tau - b}{\rho}$.

Equation (4) is at least $b$ when $\phi \geq \frac{\tau + b}{\rho}$, and so the conditions in Theorem 4.8(b) are met for any $\tilde{\phi} \geq \frac{\tau + b}{\rho}$.

## A.5   PROOF OF THEOREM 4.11

We first given the proof of Lemma 4.10.

*Proof.* Take $p^\alpha(\mathbf{y}|\mathbf{x}, \boldsymbol{\theta}^\star)$ to be the likelihood of $\mathbf{y}$ under the $Q_{\boldsymbol{\Psi}}$-mixed prior $P_{\boldsymbol{\Psi}|\boldsymbol{\theta}^\star}^\alpha$.

$$\mathcal{D}_{\boldsymbol{\theta}^\star}(\alpha) = \mathrm{D}_{\mathrm{KL}} \left( Q_{\mathbf{Y}|\mathbf{x}} \| P_{\mathbf{Y}|\mathbf{x},\boldsymbol{\theta}^\star}^\alpha \right)$$

$$= \int_{\mathscr{Y}} \log \left( \frac{q(\mathbf{y}|\mathbf{x})}{p^\alpha(\mathbf{y}|\mathbf{x}, \boldsymbol{\theta}^\star)} \right) q(\mathbf{y}|\mathbf{x}) \, d\mathbf{y}$$

$$= \int_{\mathscr{Y}} \log \left( \frac{\mathbb{E}_{\boldsymbol{\psi} \sim Q_{\boldsymbol{\Psi}}} [p(\mathbf{y}|\mathbf{x}, \boldsymbol{\theta}^\star, \boldsymbol{\psi})]}{\mathbb{E}_{\boldsymbol{\psi} \sim P_{\boldsymbol{\Psi}|\boldsymbol{\theta}^\star}^\alpha} [p(\mathbf{y}|\mathbf{x}, \boldsymbol{\theta}^\star, \boldsymbol{\psi})]} \right) q(\mathbf{y}|\mathbf{x}) \, d\mathbf{y}$$

$$= \int_{\mathscr{Y}} \log \left( \frac{\mathbb{E}_{\boldsymbol{\psi} \sim Q_{\boldsymbol{\Psi}}} [p(\mathbf{y}|\mathbf{x}, \boldsymbol{\theta}^\star, \boldsymbol{\psi})]}{\alpha \, \mathbb{E}_{\boldsymbol{\psi} \sim Q_{\boldsymbol{\Psi}}} [p(\mathbf{y}|\mathbf{x}, \boldsymbol{\theta}^\star, \boldsymbol{\psi})] + (1-\alpha) \, \mathbb{E}_{\boldsymbol{\psi} \sim P_{\boldsymbol{\Psi}|\boldsymbol{\theta}^\star}} [p(\mathbf{y}|\mathbf{x}, \boldsymbol{\theta}^\star, \boldsymbol{\psi})]} \right) q(\mathbf{y}|\mathbf{x}) \, d\mathbf{y}$$

$$= - \int_{\mathscr{Y}} \log \left( \alpha + (1-\alpha) \frac{\mathbb{E}_{\boldsymbol{\psi} \sim P_{\boldsymbol{\Psi}|\boldsymbol{\theta}^\star}} [p(\mathbf{y}|\mathbf{x}, \boldsymbol{\theta}^\star, \boldsymbol{\psi})]}{\mathbb{E}_{\boldsymbol{\psi} \sim Q_{\boldsymbol{\Psi}}} [p(\mathbf{y}|\mathbf{x}, \boldsymbol{\theta}^\star, \boldsymbol{\psi})]} \right) q(\mathbf{y}|\mathbf{x}) \, d\mathbf{y}$$

$$= - \int_{\mathscr{Y}} \log \left( \alpha + (1-\alpha) \frac{p(\mathbf{y}|\mathbf{x}, \boldsymbol{\theta}^\star)}{q(\mathbf{y}|\mathbf{x})} \right) q(\mathbf{y}|\mathbf{x}) \, d\mathbf{y}$$

$$\geq - \log \left( \int_{\mathscr{Y}} \left( \alpha + (1-\alpha) \frac{p(\mathbf{y}|\mathbf{x}, \boldsymbol{\theta}^\star)}{q(\mathbf{y}|\mathbf{x})} \right) q(\mathbf{y}|\mathbf{x}) \, d\mathbf{y} \right) \quad \text{(Jensen's inequality)}$$

$$= - \log \left( \alpha + (1-\alpha) \left( \int_{\mathscr{Y}} \frac{p(\mathbf{y}|\mathbf{x}, \boldsymbol{\theta}^\star)}{q(\mathbf{y}|\mathbf{x})} q(\mathbf{y}|\mathbf{x}) \, d\mathbf{y} \right) \right) \tag{5}$$

$\square$

For the proof of Theorem 4.8, we can without loss of generality take $\alpha_1 = 0$ and $\alpha_2 \in (0, 1]$. (Notice that for any $P_{\boldsymbol{\Psi}|\boldsymbol{\theta}^\star}$ and $\alpha_1 > 0$ we could take the prior $P_{\boldsymbol{\Psi}|\boldsymbol{\theta}^\star}^{\alpha_1} = \alpha_1 Q_{\boldsymbol{\Psi}} + (1 - \alpha_1) P_{\boldsymbol{\Psi}|\boldsymbol{\theta}^\star}$ $Q_{\boldsymbol{\Psi}}$-mixed at rate 0, which would be equivalent to $P_{\boldsymbol{\Psi}|\boldsymbol{\theta}^\star}$ $Q_{\boldsymbol{\Psi}}$-mixed at rate $\alpha_1$.) When $\alpha = 0$, we can use the usual notation for the prior and bias terms.

By Proposition 4.1, we can say that if $P_{\Psi|\mathbf{x}}$ induces negative interference under the given DGP:

$$\mathcal{D}_{\boldsymbol{\theta}^\star} > \mathcal{D}$$

$$-\mathcal{D}_{\boldsymbol{\theta}^\star} < -\mathcal{D}$$

$$-\int_{\mathscr{Y}} \log\left(\frac{q(\mathbf{y}|\mathbf{x})}{p(\mathbf{y}|\mathbf{x}, \boldsymbol{\theta}^\star)}\right) q(\mathbf{y}|\mathbf{x})\, d\mathbf{y} < -\mathcal{D}$$

$$-\log\left(\int_{\mathscr{Y}} \frac{q(\mathbf{y}|\mathbf{x})}{p(\mathbf{y}|\mathbf{x}, \boldsymbol{\theta}^\star)}\, q(\mathbf{y}|\mathbf{x})\, d\mathbf{y}\right) < -\mathcal{D} \qquad \text{(Jensen's inequality)}$$

$$\log\left(\int_{\mathscr{Y}} \frac{p(\mathbf{y}|\mathbf{x}, \boldsymbol{\theta}^\star)}{q(\mathbf{y}|\mathbf{x})}\, q(\mathbf{y}|\mathbf{x})\, d\mathbf{y}\right) < -\mathcal{D}$$

$$\int_{\mathscr{Y}} \frac{p(\mathbf{y}|\mathbf{x}, \boldsymbol{\theta}^\star)}{q(\mathbf{y}|\mathbf{x})}\, q(\mathbf{y}|\mathbf{x})\, d\mathbf{y} < e^{-\mathcal{D}}$$

$$< 1$$

The last line follows since $e^{-\mathcal{D}} \geq 1$ would violate the non-negativity of the Kullback-Leibler divergence measure that defines $\mathcal{D}$.

Since $\int_{\mathscr{Y}} \frac{p(\mathbf{y}|\mathbf{x}, \boldsymbol{\theta}^\star)}{q(\mathbf{y}|\mathbf{x})}\, q(\mathbf{y}|\mathbf{x})\, d\mathbf{y} < 1$, we can say that if $P_{\Psi|\mathbf{x}}$ induces negative interference under the given DGP, the expression in line 5 generally decreases with $\alpha$. Comparison between $\alpha = 0$ and $\alpha \in (0, 1]$ recovers the statement in the theorem for $\alpha_1 = 0$, which can be generalized to all $\alpha_1 \in [0, 1)$ and $\alpha_2 \in (0, 1] > \alpha_1$ as described above.

# B   DETAILS OF ILLUSTRATIVE EXAMPLES

## B.1   LINEAR MODEL

Data was generated according to the model $\mathbf{y} \sim \mathcal{N}\left(\boldsymbol{\theta}\mathbf{x}_1 + \boldsymbol{\psi}_1\mathbf{x}_2 + \boldsymbol{\psi}_2\mathbf{x}_3 + \boldsymbol{\psi}_3\mathbf{x}_4, \sigma^2\right)$ where $\sigma^2 = 1$. The learner's prior over $(\boldsymbol{\Theta}, \boldsymbol{\Psi}_1, \boldsymbol{\Psi}_2, \boldsymbol{\Psi}_3)$ is $\mathcal{N}(\boldsymbol{\mu}, \boldsymbol{\Sigma})$ where $\boldsymbol{\mu} = \begin{bmatrix} 0 & 0 & 0 & 0 \end{bmatrix}$ and $\boldsymbol{\Sigma} = \mathrm{diag}\left(\begin{bmatrix} 10 & 10 & 10 & 10 \end{bmatrix}\right)$.

We use $\sigma^2_{\mathbf{y}|\mathbf{x}}$ and $\sigma^2_{\mathbf{y}|\mathbf{x}, \boldsymbol{\theta}^\star}$ to refer to the variance of the learner's predictive distribution and variance of distribution corresponding to the learner's target likelihood, respectively. These are:

$$\sigma^2_{\mathbf{y}|\mathbf{x}} = \sigma^2 + \mathbf{x}\boldsymbol{\Sigma}\mathbf{x}^T \tag{6}$$

$$\sigma^2_{\mathbf{y}|\mathbf{x}, \boldsymbol{\theta}^\star} = \sigma^2 + \mathbf{x} \begin{bmatrix} 0 & 0 \\ 0 & \boldsymbol{\Sigma}_{(2:4),(2:4)} - \boldsymbol{\Sigma}_{1,(2:4)}\boldsymbol{\Sigma}_1^{-1}\boldsymbol{\Sigma}_{1,(2:4)} \end{bmatrix} \mathbf{x}^T \tag{7}$$

We used the following formulas for the ETIG and ELIG:

$$\mathrm{ETIG}(\mathbf{x}) = \frac{1}{2}\log\left(\frac{\sigma^2_{\mathbf{y}|\mathbf{x}}}{\sigma^2_{\mathbf{y}|\mathbf{x}, \boldsymbol{\theta}^\star}}\right) \tag{8}$$

$$\mathrm{ELIG}(\mathbf{x}) = \frac{1}{2}\left(\log\left(\frac{\sigma^2_{\mathbf{y}|\mathbf{x}}}{\sigma^2}\right) - \log\left(\frac{\sigma^2_{\mathbf{y}|\mathbf{x}}}{\sigma^2_{\mathbf{y}|\mathbf{x}, \boldsymbol{\theta}^\star}}\right)\right) \tag{9}$$

To generate the set of possible actions, we sampled 10,000 values $\mathbf{z} \sim \mathcal{N}(10, .25)$. For each value of $\mathbf{z}$, we then sampled one value of each $\mathbf{x}_2$, $\mathbf{x}_3$ and $\mathbf{x}_4$ from $\mathcal{N}(\mathbf{z}, .25)$, and one value of $\mathbf{x}_1$ from $\mathcal{N}(-1/\mathbf{z}, .25)$. Each point in Figure 1b corresponds to one of 10,000 values of $\mathbf{x}$ generated as above. With reference to Figure 1a, when we calculate $r^\star(\mathbf{x})$, $\mathbf{x}$ is always $\mathrm{argmax}_{\mathbf{x} \in \mathscr{X}} \mathrm{ETIG}(\mathbf{x})$ where $\mathscr{X}$ contains all 10,000 possible actions.

## B.2   PREFERENCE MODELING

We modified the preference example from Foster et al. [2019], who use a censored sigmoid normal as the output distribution. Instead, we used the Bernoulli distribution $\mathbf{y} \sim \mathrm{Bernoulli}\left(\frac{1}{1+e^{\psi\mathbf{x}-\theta}}\right)$.

Figure 1c shows the $r^\star(\mathbf{x})$ for $\mathbf{x} = \text{argmax}_{\mathbf{x} \in \mathscr{X}} \text{ETIG}(\mathbf{x})$ where $\mathscr{X}$ contains values evenly spaced between -79 and 81 (this differs slightly from the example in Foster et al. [2019], in which values of $\mathbf{x}$ were evenly spaced between -80 and 80).

Unlike in the linear model setting, for which there are closed-form expressions for the ETIG and ELIG, the ETIG and ELIG for this example are not known in closed form. We approximate them using the following nested Monte Carlo estimators:

$$\widehat{\text{ETIG}}(\mathbf{x}) = \sum_{i=1}^{N} \left( \sum_{\mathbf{y} \in \mathscr{Y}} \left( p(\mathbf{y}|\mathbf{x}, \boldsymbol{\theta}^i, \boldsymbol{\psi}^i) \log \left( \frac{\sum_{j=1}^{M} p(\mathbf{y}|\mathbf{x}, \boldsymbol{\theta}^i, \boldsymbol{\psi}^j)}{\sum_{l=1}^{N} p(\mathbf{y}|\mathbf{x}, \boldsymbol{\theta}^l, \boldsymbol{\psi}^l)} \right) \right) \right) \tag{10}$$

$$\widehat{\text{ELIG}}(\mathbf{x}) = \sum_{i=1}^{N} \left( \sum_{\mathbf{y} \in \mathscr{Y}} \left( p(\mathbf{y}|\mathbf{x}, \boldsymbol{\theta}^i, \boldsymbol{\psi}^i) \log \left( \frac{p(\mathbf{y}|\mathbf{x}, \boldsymbol{\theta}^i, \boldsymbol{\psi}^i)}{\sum_{l=1}^{N} p(\mathbf{y}|\mathbf{x}, \boldsymbol{\theta}^l, \boldsymbol{\psi}^l)} \right) \right) \right) - \widehat{\text{ETIG}}(\mathbf{x}) \tag{11}$$

Samples of $\boldsymbol{\theta}$ and $\boldsymbol{\psi}$ are drawn from the prior given above. Although not shown explicitly in Equation (10), each set of $M$ inner samples is constrained to include the corresponding sample $(\boldsymbol{\theta}^i, \boldsymbol{\psi}^i)$ to avoid pathological behavior when a value $\mathbf{y}$ has positive probability in only a very small region of $P_{\boldsymbol{\Psi}|\boldsymbol{\theta}^i}$ [Foster et al., 2020]. We set $N$ to 10,000 and $M$ to 100 (reflecting results from Rainforth et al. [2018] that $M$ is optimally $\propto \sqrt{N}$).

*Remark on Figure 1d.* The trade-off between ETIG and ELIG visualized in Figure 1d can be explained by noticing that the magnitude of $\mathbf{x}$ has opposite effects on the ease of identification of $\boldsymbol{\theta}$ and $\boldsymbol{\psi}$. When $\mathbf{x} = 0$, $\boldsymbol{\psi}\mathbf{x} = 0$ and so $\boldsymbol{\psi}$ can't be identified at all. This is reflected by the fact that the ETIG peaks at 0. Conversely, the size of the effect of $\boldsymbol{\psi}$ on outcomes depends on the magnitude of $\mathbf{x}$, which is reflected by the fact that the ELIG is maximized by values of $\mathbf{x}$ with large magnitudes.

## B.3 GAUSSIAN PROCESS REGRESSION

We constructed the kernel $k(\cdot, \cdot)$ as the additive composition of $k_{\boldsymbol{\theta}}(\cdot, \cdot)$ and $k_{\boldsymbol{\psi}_1}(\cdot, \cdot)$ where both $k_{\boldsymbol{\theta}}(\cdot, \cdot)$ and $k_{\boldsymbol{\psi}_1}(\cdot, \cdot)$ were radial basis functions kernels with shared amplitude and lengthscale determined by the values of $\boldsymbol{\theta}$ and $\boldsymbol{\psi}_1$, respectively.

To generate Figure 1e, we set $\mathbf{x} = [25, 26]$, and sampled 10,000 values from the learner's joint distribution over $(\boldsymbol{\theta}, \boldsymbol{\psi}_1, \boldsymbol{\psi}_2)$, where $\boldsymbol{\psi}_2$ is a sample from the GP at $\mathbf{x}$. We set $\sigma^2$, the variance of $\mathbf{Y}|\mathbf{x}, \boldsymbol{\theta}, \boldsymbol{\psi}_1, \boldsymbol{\psi}_2$, to .01.

We approximated $r^\star(\mathbf{x})$ using a nested Monte Carlo estimator:

$$r^\star(\mathbf{x}) = \mathop{\mathbb{E}}_{\mathbf{y} \sim P_{\mathbf{Y}|\boldsymbol{\theta}^\star, \boldsymbol{\psi}^\star}} \left[ \log\left( p(\mathbf{y}|\mathbf{x}, \boldsymbol{\theta}^\star) \right) - \log\left( p(\mathbf{y}|\mathbf{x}) \right) \right]$$

$$\approx \frac{1}{M} \sum_{i=1}^{M} \left( \log \left( \frac{1}{M} \sum_{j=1}^{M} p(y^i|\mathbf{x}, \boldsymbol{\theta}^\star, \boldsymbol{\psi}^j) \right) - \log \left( \frac{1}{N} \sum_{j=1}^{N} p(y^i|\mathbf{x}, \boldsymbol{\theta}^j, \boldsymbol{\psi}^j) \right) \right)$$

with $N = 10,000$ and $M = 100$. Samples were drawn from the prior given above.

To generate Figure 1f, we used a Hamiltonian Monte Carlo (HMC) sampler [Gardner et al., 2018] to first train the model, initialized with the priors given above, on the five randomly-sampled points shown in the figure. The training data was generated from a function sampled from a kernel with $\boldsymbol{\theta}^\star = 5$ and $\boldsymbol{\psi}_1^\star = 2.5$. In this case, $\sigma^2 = 0$, i.e., outcomes were treated as deterministic. We again used nested Monte Carlo estimators for the acquisition functions, given below. Rather than sampling from the prior, we used the HMC samples of $\boldsymbol{\theta}$ and $\boldsymbol{\psi}_1$ to compute the expectations.

$$\text{ETIG}(\mathbf{x}) = \mathop{\mathbb{E}}_{\boldsymbol{\theta}, \boldsymbol{\psi}, \mathbf{y} \sim P_{\boldsymbol{\Theta}, \boldsymbol{\Psi}, \mathbf{Y}|\mathbf{x}}} \left[ \log\left( p(\mathbf{y}|\mathbf{x}, \boldsymbol{\theta}) \right) - \log\left( p(\mathbf{y}|\mathbf{x}) \right) \right]$$

$$\approx \frac{1}{N} \sum_{i=1}^{N} \left( \frac{1}{L} \sum_{l=1}^{L} \log \left( \frac{1}{M} \sum_{j=1}^{M} p(\mathbf{y}^l|\mathbf{x}, \boldsymbol{\theta}^i, \boldsymbol{\psi}^j) \right) - \frac{1}{L} \sum_{l=1}^{L} \log \left( \frac{1}{N} \sum_{j=1}^{N} p(\mathbf{y}^l|\mathbf{x}, \boldsymbol{\theta}^j, \boldsymbol{\psi}^j) \right) \right)$$

$$\text{ELIG}(\mathbf{x}) = \underset{\boldsymbol{\theta},\boldsymbol{\psi},\mathbf{y} \sim P_{\boldsymbol{\Theta},\boldsymbol{\Psi},\mathbf{Y}|\mathbf{x}}}{\mathbb{E}} \left[ \log\left(p(\mathbf{y}|\mathbf{x},\boldsymbol{\theta},\boldsymbol{\psi})\right) - \log\left(p(\mathbf{y}|\mathbf{x})\right) \right] - \underset{\boldsymbol{\theta},\boldsymbol{\psi},\mathbf{y} \sim P_{\boldsymbol{\Theta},\boldsymbol{\Psi},\mathbf{Y}|\mathbf{x}}}{\mathbb{E}} \left[ \log\left(p(\mathbf{y}|\mathbf{x},\boldsymbol{\theta})\right) - \log\left(p(\mathbf{y}|\mathbf{x})\right) \right]$$

$$= \underset{\boldsymbol{\theta},\boldsymbol{\psi},\mathbf{y} \sim P_{\boldsymbol{\Theta},\boldsymbol{\Psi},\mathbf{Y}|\mathbf{x}}}{\mathbb{E}} \left[ \log\left(p(\mathbf{y}|\mathbf{x},\boldsymbol{\theta},\boldsymbol{\psi})\right) \right] - \underset{\boldsymbol{\theta},\boldsymbol{\psi},\mathbf{y} \sim P_{\boldsymbol{\Theta},\boldsymbol{\Psi},\mathbf{Y}|\mathbf{x}}}{\mathbb{E}} \left[ \log\left(p(\mathbf{y}|\mathbf{x},\boldsymbol{\theta})\right) \right]$$

$$= - \underset{\boldsymbol{\theta},\boldsymbol{\psi},\mathbf{y} \sim P_{\boldsymbol{\Theta},\boldsymbol{\Psi},\mathbf{Y}|\mathbf{x}}}{\mathbb{E}} \left[ \log\left(p(\mathbf{y}|\mathbf{x},\boldsymbol{\theta})\right) \right]$$

$$\approx -\frac{1}{N}\sum_{i=1}^{N} \left( \frac{1}{L}\sum_{l=1}^{L} \log\left( \frac{1}{M}\sum_{j=1}^{M} p(\mathbf{y}^l|\mathbf{x},\boldsymbol{\theta}^i,\boldsymbol{\psi}^j) \right) \right)$$

with $N = M = 500$ and $L = 10,000$. (Since observations are treated as deterministic, the term in the ELIG corresponding to the entropy of $\mathbf{Y}|\boldsymbol{\theta},\boldsymbol{\psi}$ is 0.)