# OpenReview forum: "Bayesian Active Learning in the Presence of Nuisance Parameters"
_auai.org/UAI/2024/Conference — UAI 2024 oral_

### Official Review · Reviewer_jDzS · 2024-03-22

**Q2-1 Originality-Novelty:** 2
**Q2-2 Correctness-Technical Quality:** 3
**Q2-5 Clarity Of Writing:** 3

**Q1 Summary And Contributions:**

This paper  introudces  a  theoretical formulation about how Bayesian active learning is affected by misspecification of nuisance parameters, called negative interference, and provide conditions under which negative interference arises and may be large.

**Q2-3 Extent To Which Claims Are Supported By Evidence:**

3: Good: the main claims are supported by convincing evidence (in the form of adequate experimental evaluation, proofs, (pseudo-)code, references, assumptions).

**Q2-4 Reproducibility:**

3: Good: key resources (e.g. proofs, code, data) are available and key details (e.g. proofs, experimental setup) are sufficiently well-described for competent researchers to confidently reproduce the main results.

**Q3 Main Strengths:**

1. They give a novel theoretical formulation about how Bayesian active learning is affected by misspecification of nuisance parameters, called negative interference, and provide conditions under which negative interference arises and may be large.
2. They propose a new framework that active learner should trade between aquisition functions of primary and auxiliary objectives. The acquisition function should not only depend on information gain for the primary objective, otherwise performance may be drastically bad.

**Q4 Main Weakness:**

1. Their results are not so surprising. Moreover, they only propose the problem about the active learner's dilemma, but not propose some methodologies for this problem.
2. Their setting is not general enough to cover most settings of Bayesian active learning, such as Bayesian optimization where target is to maximize the outcome.

**Q5 Detailed Comments To The Authors:**

Theorem 4.8 says that there always exists a sequence of Nuisance parameter error a priori settings such that $r^*(x)$ tends to negative infinity. It seems to say that we must optimize the Nuisance parameter simultaneously. But in real applications, this prior is given, so it is not necessarily true that we must optimize the Nuisance parameter. I have a question, can the necessity of optimizing the Nuisance parameter be learned?

**Q9 Complying With Reviewing Instructions:**

Yes

---

> ### Author Rebuttal · Authors · 2024-04-04
>
> *Their setting is not general enough to cover most settings of Bayesian active learning, such as Bayesian optimization where target is to maximize the outcome.*
>
> We respectfully disagree. Our formulation encompasses settings like Bayesian optimization as a special case. In particular, in Bayesian optimization, the target parameter is the location of the data-generating function maximum, and the nuisance parameters correspond to the parameters of the data-generating function.
>
> *But in real applications, this prior is given, so it is not necessarily true that we must optimize the Nuisance parameter.*
>
> The premise of our work is that the prior one is given may differ from the true distribution of nuisance parameters (i.e., be misspecified). You are correct that our results pertaining to mitigation of the threat of negative interference will be less easily applied in a situation where this prior is not updated.
>
> *[C]an the necessity of optimizing the Nuisance parameter be learned?*
>
> **We agree that this is an important question, and will add discussion of how to achieve this to Sec. 6.** The question of whether it is possible to detect if one is in the region of positive or negative interference is critical for the practical application of our results. We will add the following to Sec. 6 (see also our comment above entitled "Author Rebuttal"): "Applications of sOED in [the fixed-$\boldsymbol{\psi}$ formulation] can address the threat of negative interference with design policies that alternate between the ELIG and ETIG according to a principled switching criterion (one that would ideally be sensitive to the extent of the threat of negative interference). This criterion could be based on one of the many existing criteria developed in the context of the exploration-exploitation dilemma (e.g., an $\epsilon$-greedy scheme), the value of a distributionally robust information gain measure (Go and Isaac, 2022), or a credible lower bound of target information gain values."

---

### Official Review · Reviewer_YZ77 · 2024-03-22

**Q2-1 Originality-Novelty:** 4
**Q2-2 Correctness-Technical Quality:** 4
**Q2-5 Clarity Of Writing:** 3

**Q10 Ethical Concerns:**

No.

**Q1 Summary And Contributions:**

The paper studies the concept of interference, defined as the effect nuisance parameters have on the information gain between actions and target parameters. It asks three questions: when does interference by nuisance parameters occur, how large is the interference effect and how can an algorithm mitigate interference? These are addressed in detail through a series of analyses and illustrated with several experiments.

**Q2-3 Extent To Which Claims Are Supported By Evidence:**

3: Good: the main claims are supported by convincing evidence (in the form of adequate experimental evaluation, proofs, (pseudo-)code, references, assumptions).

**Q2-4 Reproducibility:**

3: Good: key resources (e.g. proofs, code, data) are available and key details (e.g. proofs, experimental setup) are sufficiently well-described for competent researchers to confidently reproduce the main results.

**Q3 Main Strengths:**

- The paper clearly and precisely formulates the problem of interest, with connections to other relevant problem settings such as active model selection, preference learning and transfer learning.
- The paper addresses key research questions with thorough analyses.
- Theorems are also explained in plain English and with simple examples.
- The illustrative examples show that negative interference is not confined to pathological settings.

**Q4 Main Weakness:**

- Proofs and key details of illustrative examples are deferred to the Supplementary files instead of being included in the paper.

**Q5 Detailed Comments To The Authors:**

- (Sec. 4) The explanation of the example tied to Eq. 1 is incomplete. It covers the effect of prior misspecification on an inference process over $\theta^*$ but doesn't mention how $P_\Psi,\Theta$ differs from $Q_\Psi$.
- (Sec. 5) For the linear model, preference learning and GP regression examples, I think it's important to at least mention the prior specification in the main text. For space, I recommend removing some references.
- (App. B) In B.1, the ETIG and second term of ELIG contain a term with $xSx^{\top}$ in a numerator and $x (S_{(2:4),(2:4)} .. ) x^{\top}$ in the denominator. The matrix has a different size in the numerator (4 by 4) than in the denominator (3 by 3). $x$ is 4-dimensional, so the denominator seems to be incorrect or confusingly noted.

**Q9 Complying With Reviewing Instructions:**

Yes

---

> ### Author Rebuttal · Authors · 2024-04-04
>
> *(Sec. 4).*
>
> We will replace the text below Eq. 1 with the following:
> "Consider a learner to whom $\sigma^2$ is known, and who assigns to $(\boldsymbol{\Theta}, \boldsymbol{\Psi})$ a normal prior, i.e., $P_{\boldsymbol{\Theta}, \boldsymbol{\Psi}} = \mathcal{N}([\mu_\theta, \mu_\psi], \mathrm{diag}(s_\theta, s_\psi))$. The learner may be mistaken about the distribution of $\boldsymbol{\Psi}$, and the degree to which they are mistaken will affect their estimate of $\boldsymbol{\theta}$. Informally, if $Q_{\boldsymbol{\Psi}}$ favors large values, the learner will tend to observe large values of $\boldsymbol{y}$. However, if the learner's prior $P_{\boldsymbol{\Psi}}$ favors small values, the learner will infer that the large values of $\mathbf{y}$ they observe are due to a large value of $\boldsymbol{\theta}$. Importantly, this will occur regardless of the true value of $\boldsymbol{\theta}$ - and so if $\boldsymbol{\theta}$ is relatively small, the result will be negative interference. In this example, the more mistaken the learner is about the distribution of $\boldsymbol{\Psi}$, the more they will misattribute their observations to the value of $\boldsymbol{\theta}$. If they are arbitrarily mistaken, the amount of negative interference can become arbitrarily large. In other words, $r^\star(\boldsymbol{x})$ does not have a finite lower bound.
>
> To make this intuition more concrete, consider a simple case where $Q_{\boldsymbol{\Psi}}$ is known to be Gaussian but need not be centered at $\mu_\psi$ (for the sake of exposition, assume $s_\psi$ is known to the learner). Possible values of the mean of $Q_{\boldsymbol{\Psi}}$ can be captured by all numbers on the real line $\mathbb{R}$, which index a set of possible distributions $\mathscr{F} = \\{ \mathcal{N}(\boldsymbol{\phi}, s_\psi) ~ \vert ~ \boldsymbol{\phi} \in \mathbb{R} \\}$. Since $\mathbb{R}$ is unbounded, $\mathscr{F}$ satisfies Assumption 4.7. The intuition established above can now be stated more formally as '$r^\star(\boldsymbol{x})$ will decrease with the magnitude of $\boldsymbol{\phi}$ (as it moves further from $\mu_\psi$); since there is no limit to the magnitude of $\boldsymbol{\phi}$, there is no limit to the extent of negative interference.' We show in Appendix A.4 that this intuition holds in the sense that this simple case satisfies both conditions of Theorem 4.8." (In line with a suggestion made by Reviewer FVbE to have a simple running example for exposition, we actually plan to additionally move much of the first paragraph of the preceding text to just before the start of Sec. 3.)
>
> *(Sec. 5).*
>
> We agree that this is important information and will mention the prior specification for the experiments in the main text.
>
> *(App. B).*
>
> **You are correct in pointing out this error, and we will correct it**; thank you for your careful reading. The matrix $S\_{(2:4),(2:4)} - S\_{1,(2:4)} S^{-1}\_1 S\_{1,(2:4)}$ is a submatrix of the covariance matrix $S$ conditioned on a particular value $\boldsymbol{\theta}$. The $3 \times 3$ matrix captures the remaining uncertainty about the values of $\boldsymbol{\psi}_1$, $\boldsymbol{\psi}_2$ and $\boldsymbol{\psi}_3$, and is the bottom right submatrix of the entire posterior covariance matrix (the other entries of the entire $4 \times 4$ matrix are 0, since there is no remaining uncertainty about the value of $\boldsymbol{\theta}$). As you correctly point out, these equations should instead contain the entire posterior covariance matrix.

---

### Official Review · Reviewer_FVbE · 2024-03-23

**Q2-1 Originality-Novelty:** 2
**Q2-2 Correctness-Technical Quality:** 3
**Q2-5 Clarity Of Writing:** 2

**Q1 Summary And Contributions:**

This paper shows that the existence of nuisance parameters can pose a problem for Bayesian active learning, as the learner may pick samples believed to be informative about the target parameter, but that actually make the posterior move away from the true value of the target parameter. This problem, called negative interference, is studied through several theoretical results and a number of synthetic examples.

**Q2-3 Extent To Which Claims Are Supported By Evidence:**

3: Good: the main claims are supported by convincing evidence (in the form of adequate experimental evaluation, proofs, (pseudo-)code, references, assumptions).

**Q2-4 Reproducibility:**

3: Good: key resources (e.g. proofs, code, data) are available and key details (e.g. proofs, experimental setup) are sufficiently well-described for competent researchers to confidently reproduce the main results.

**Q3 Main Strengths:**

The concepts introduced are useful, and their analysis is sound from what I can see.

(I am not an expert on active learning, so I can't say much about the comparison to related work.)

**Q4 Main Weakness:**

An algorithm (even a fairly naive one) built on these insights would have been a nice addition to flesh out the contribution.

I think some changes in the structure of the paper would help to make it more readable. After the claims in the introduction, it takes fairly long until they are made precise; the related work section in between only added to the questions in my mind about the specifics of what will be shown in this paper. Section 4 could be easier to navigate. Having an example from section 5 earlier in the paper (e.g. in the introduction) would be helpful to make more precise early on what the problem is. I also have some concerns (discussed below) about appropriateness of notation used, which makes some aspects of the discussion less transparent.

**Q5 Detailed Comments To The Authors:**

**Notation for priors and posteriors:** $P$ and $p$ are used to describe distributions and probabilities known to the learner, including ones that are given (like $p(y | x,\theta,\psi)$ and ones that need to be learned. Distinguishing between these two (and further, between priors and posteriors) could provide several advantages. First, at the top right of page 2, it is specified how the posterior over $\theta$ is computed. By not mentioning the nuisance parameters $\psi$, this leaves open the question whether the prior on $\psi$ is also being updated. (I assume it is - though in that case the "target likelihood" $p(y|x,\theta^*$ varies over time, and it seems odd that you would define a moving target.) Then in definition 2.1, the notation $I(P_\Theta; P_{Y|x})$ is used for mutual information, according to footnote 2 to make clear that both distributions are w.r.t. the prior. However, mutual information is not a property of two separate distributions, but of two *random variables* (or equivalently, of the *joint* distribution of those variables). Explicit notation for the Bayesian random variables describing the learner's beliefs about the parameters would be a better solution. Distinguishing between prior and posterior (/ posterior after $n$ samples) would also allow discussion of the question what happens after a *sequence* of samples: Can negative interference continue, or will it improve after some number of samples even if the learner is focusing only on the target parameters?

Other comments:
- Below assumption 4.4, it is suggested to take $\epsilon=0$ in the discrete case, but this would make $N_\epsilon(\cdot) = \varnothing$.
- Section 5, first setting: "negatively and inversely correlated" - These are synonyms, as correlations are a concept based on linear relations and cannot express what you mean by "inversely".
- Claim about prior misspecification: "Firstly, rather than characterizing pathological edge cases, a substantial proportion of DGPs induce negative interference." - Can you justify drawing this conclusion from three examples of DGPs that were (at least partially) constructed for the purpose of illustrating negative interference?
- Caption of figure 1, about (b): "Rescaling" usually means multiplying by a nonzero constant, which wouldn't allow getting the minima to fall at 0. Maybe "translating" better describes the operation you performed?

**Q9 Complying With Reviewing Instructions:**

Yes

---

> ### Author Rebuttal · Authors · 2024-04-04
>
> *Having an example... would be helpful to make more precise early on what the problem is.*
>
> We will move part of the example containing Eq. 1 to just before the start of Sec. 3, alongside the following explanation: "Consider a learner to whom $\sigma^2$ is known, and who assigns to $(\boldsymbol{\Theta}, \boldsymbol{\Psi})$ a normal prior, i.e., $P_{\boldsymbol{\Theta}, \boldsymbol{\Psi}} = \mathcal{N}([\mu_\theta, \mu_\psi], \mathrm{diag}(s_\theta, s_\psi))$. The learner may be mistaken about the distribution of $\boldsymbol{\Psi}$, and the degree to which they are mistaken will affect their estimate of $\boldsymbol{\theta}$. Informally, if $Q_{\boldsymbol{\Psi}}$ favors large values, the learner will tend to observe large values of $\boldsymbol{y}$. However, if the learner's prior $P_{\boldsymbol{\Psi}}$ favors small values, the learner will infer that the large values of $\mathbf{y}$ they observe are due to a large value of $\boldsymbol{\theta}$. Importantly, this will occur regardless of the true value of $\boldsymbol{\theta}$ - and so if $\boldsymbol{\theta}$ is relatively small, the result will be negative interference." We will maintain this running example by also including discussion on the application of Thm. 4.8 to the example after stating the theorem.
>
> *Notation for priors and posteriors.*
>
> **We will add to our exposition of the general formulation in Sec. 2 to clarify the following points about our notation.** Our choices reflect our goal of establishing a maximally general formulation. For example, you noted that "[b]y not mentioning the nuisance parameters $\boldsymbol{\psi}$, this leaves open the question whether the prior on $\boldsymbol{\psi}$ is also being updated." You are absolutely correct; this omission is intentional. Our theoretical results do not depend on whether the prior on $\boldsymbol{\psi}$ is also being updated (or on whether the prior is actually the posterior after $n$ samples), and so our formulation allows for both. As we acknowledge at the beginning of Sec. 2, we agree that ambiguity is introduced by our choice to distinguish between RVs that are un/available to the learner only via the notation for their corresponding probability distributions (we made this choice in order to minimize the amount of unfamiliar notation introduced). To add additional clarity, we will write the mutual information as $I(\boldsymbol{\Theta} \sim P_{\boldsymbol{\Theta}} ~ ; ~ \boldsymbol{Y} \sim P_{\boldsymbol{Y} \vert \boldsymbol{x}})$.
>
> *Can negative interference continue, or will it improve after some number of samples even if the learner is focusing only on the target parameters?*
>
> Negative interference can continue indefinitely in some cases (see related discussion in the first paragraph on pg. 7, just before the paragraph titled "Connection to negative transfer").
>
> *Below assumption 4.4, it is suggested to take $\epsilon = 0$ in the discrete case, but this would make $N_\epsilon(\cdot) = \varnothing$.*
>
> Thank you for pointing this out. We believe the statement can be rigorously interpreted in the following way (which for clarity, we will add to the text): As $\epsilon$ is the maximum distance of elements of $N_\epsilon(\boldsymbol{\theta}^\star)$ from $\boldsymbol{\theta}^\star$, if $\epsilon = 0$, $N_\epsilon(\boldsymbol{\theta}^\star) = \\{ \boldsymbol{\theta}^\star \\}$ since only $\boldsymbol{\theta}^\star$ is distance 0 from itself.
>
> *"[N]egatively and inversely correlated".*
>
> We intended to convey that "$\boldsymbol{x_1}$... is negatively correlated with the inverse of $\boldsymbol{x_{(2:4)}}$", and will correct this.
>
> *Claim about prior misspecification.*
>
> We intended to convey that "in each of these illustrative settings, a substantial proportion of the considered DGPs (i.e., unique values of $(\boldsymbol{\theta}^\star,\boldsymbol{\psi}^\star)$) induce negative interference", and will include this clarification in the text.
>
> *Caption of figure 1.*
>
> Thank you for your careful reading. We will correct this to "translating".

---

### Official Review · Reviewer_sJp9 · 2024-03-25

**Q2-1 Originality-Novelty:** 3
**Q2-2 Correctness-Technical Quality:** 2
**Q2-5 Clarity Of Writing:** 2

**Q1 Summary And Contributions:**

The paper considers the problem of active learning in the presence of nuisance parameters. In many scientific problems, models contain many parameters that affect the data distribution but are not of interest to the modeller. The standard Bayesian solution is to integrate out the nuisance parameters to obtain the distribution of parameters of interest. The authors investigate the performance of this strategy in the active setting and show that if the prior is misspecified, the active learner can be misled away from the true values of the parameters. The authors prove that under certain conditions, the magnitude of this error is unbounded and therefore impossible to overcome no matter the amount of data acquired. The authors then propose loss functions that accommodate this problem by forcing the learner to learn the values of the nuisance parameters and posit that a tradeoff exists between learning target and nuisance parameters. The authors do not provide a specific method to resolve this tradeoff, but they illustrate it using computational examples.

**Q2-3 Extent To Which Claims Are Supported By Evidence:**

3: Good: the main claims are supported by convincing evidence (in the form of adequate experimental evaluation, proofs, (pseudo-)code, references, assumptions).

**Q2-4 Reproducibility:**

3: Good: key resources (e.g. proofs, code, data) are available and key details (e.g. proofs, experimental setup) are sufficiently well-described for competent researchers to confidently reproduce the main results.

**Q3 Main Strengths:**

The underlying problem is important within Bayesian active learning and the implications of negative interference are profound, so this problem is a worthwhile avenue for research.

**Q4 Main Weakness:**

- While the paper does provide examples, the theoretical part is written in such a way that it is hard to develop intuition about the extent and consequences of negative interference.

- Some of the assumptions are informal and have to be made more precise. Assumption 4.6, which is quite important to the paper, should be restated in a rigorous way. Most distributions can be represented as vectors if we do not insist on the smoothness of the mapping between the parameters and the distribution.

- the key example following Theorem 4.8 is stated in a way that is too vague and I am not sure what the authors mean when discussing it. It should be made more explicit to serve as an intuitive example.

- While the decomposition is interesting and the problem is challenging to analyze in detail, the paper would be much more meaningful if the authors provided some guidelines/heuristics to address the tradeoff that is the main subject of the paper.

**Q5 Detailed Comments To The Authors:**

- the key example following Theorem 4.8 is stated in a way that is too vague and I am not sure what the authors mean when discussing it. It should be made more explicit to serve as an intuitive example.

**Q9 Complying With Reviewing Instructions:**

Yes

---

> ### Author Rebuttal · Authors · 2024-04-04
>
> *Assumption 4.6, which is quite important to the paper, should be restated in a rigorous way. Most distributions can be represented as vectors if we do not insist on the smoothness of the mapping between the parameters and the distribution.*
>
> ***We will revise the text to address this important point***, and thank you for raising it. For our purposes, we can restrict our analysis to distributions that can be represented as vectors, which, as you point out, applies to a broad set of distributions. To avoid unnecessary confusion, we will state explicitly that the setting for the theorem is one in which $Q_{\boldsymbol{\Psi}}$ belongs to a distribution class and can be represented as a vector, restate the assumption as a definition, and include a remark on the generality of the setting.
>
> *[T]he key example following Theorem 4.8 is stated in a way that is too vague...*
>
> We will replace the text below Eq. 1 with the following: "Consider a learner to whom $\sigma^2$ is known, and who assigns to $(\boldsymbol{\Theta}, \boldsymbol{\Psi})$ a normal prior, i.e., $P_{\boldsymbol{\Theta}, \boldsymbol{\Psi}} = \mathcal{N}([\mu_\theta, \mu_\psi], \mathrm{diag}(s_\theta, s_\psi))$. The learner may be mistaken about the distribution of $\boldsymbol{\Psi}$, and the degree to which they are mistaken will affect their estimate of $\boldsymbol{\theta}$. Informally, if $Q_{\boldsymbol{\Psi}}$ favors large values, the learner will tend to observe large values of $\boldsymbol{y}$. However, if the learner's prior $P_{\boldsymbol{\Psi}}$ favors small values, the learner will infer that the large values of $\mathbf{y}$ they observe are due to a large value of $\boldsymbol{\theta}$. Importantly, this will occur regardless of the true value of $\boldsymbol{\theta}$ - and so if $\boldsymbol{\theta}$ is relatively small, the result will be negative interference. In this example, the more mistaken the learner is about the distribution of $\boldsymbol{\Psi}$, the more they will misattribute their observations to the value of $\boldsymbol{\theta}$. If they are arbitrarily mistaken, the amount of negative interference can become arbitrarily large. In other words, $r^\star(\boldsymbol{x})$ does not have a finite lower bound.
>
> To make this intuition more concrete, consider a simple case where $Q_{\boldsymbol{\Psi}}$ is known to be Gaussian but need not be centered at $\mu_\psi$ (for the sake of exposition, assume $s_\psi$ is known to the learner). Possible values of the mean of $Q_{\boldsymbol{\Psi}}$ can be captured by all numbers on the real line $\mathbb{R}$, which index a set of possible distributions $\mathscr{F} = \\{ \mathcal{N}(\boldsymbol{\phi}, s_\psi) ~ \vert ~ \boldsymbol{\phi} \in \mathbb{R} \\}$. Since $\mathbb{R}$ is unbounded, $\mathscr{F}$ satisfies Assumption 4.7. The intuition established above can now be stated more formally as '$r^\star(\boldsymbol{x})$ will decrease with the magnitude of $\boldsymbol{\phi}$ (as it moves further from $\mu_\psi$); since there is no limit to the magnitude of $\boldsymbol{\phi}$, there is no limit to the extent of negative interference.' We show in Appendix A.4 that this intuition holds in the sense that this simple case satisfies both conditions of Theorem 4.8." (In line with a suggestion made by Reviewer FVbE to have a simple running example for exposition, we actually plan to additionally move much of the first paragraph of the preceding text to just before the start of Sec. 3.)

---

### Meta-Review · Area_Chair_C8dX · 2024-04-21

The authors consider the problem of Bayesian active learning in the presence of nuisance parameters. The challenge in this setting is that nuisance parameters can lead to bias in the Bayesian learner’s estimate of the target parameters (negative interference). The authors demonstrate the extent of negative interference can be extremely large, and that accurate estimation of the nuisance parameters is critical to reducing it. The reviewers all acknowledged the importance of the problem and the contribution of this manuscript. However, there were concerns regarding that the work only propose the problem, but not propose some methodologies for this problem and no algorithms, as well as concerns regarding presentation, especially for the assumptions.